# Understanding and Relaxing the Limitations of Transformers for Linear Algebra

**Andres Potapczynski**[*], **Alex Ali**[*] **& Andrew Gordon Wilson**
New York University
`ap6604@nyu.edu, aa10253@nyu.edu, andrewgw@cims.nyu.edu`

## Abstract

Matrix operations, such as linear solves, eigendecompositions, and log determinants, are foundational building blocks for any number of downstream applications. Therefore, any broadly capable learning system should be able to effectively approximate these operations in its internal representation. Accordingly, there is great motivation to study how transformers can perform linear algebra — for if transformers cannot even semi-competently perform matrix operations, then we cannot expect them to form a basis for a generally intelligent system. We demonstrate that current techniques developing transformers for linear algebra have striking failure modes, prohibitive scaling, and particularly poor out-of-distribution generalization to other matrix distributions, and matrices of different sizes. Investigating further, we find that current transformer approaches operate as statistical interpolators, rather than discovering algorithms that will generalize to matrices from other distributions. Based on our understanding of these limitations, we develop a sequence of interventions that substantially improve scaling and performance, including matrix embeddings through a learnable projection, linear attention, looping, and a data pre-training distribution of structured matrices. We term the resulting method the *RangeFormer*, which we show has significantly improved scaling and performance on challenging OOD matrices from the *matrix market*. Moreover, with Range-Former we show for the first time that transformers can be successfully applied to downstream tasks that involve iterative matrix operations, including Gaussian process learning, and improving the sampling distribution of randomized methods.

## 1 Introduction

Linear algebra is at the heart of science and engineering. Linear solves, eigenvalue computations, log determinants, and other matrix operations, are the basis for Gaussian processes, normalizing flows, principal component analysis, equivariant neural networks, differential equations, approximate second-order optimizers, and any number of other applications (Rasmussen and Williams, 2006; Anil et al., 2020; Cuturi, 2013; Dao et al., 2019; Finzi et al., 2023; Fu et al., 2023; Kovachki et al., 2021; Li et al., 2018; Martens and Grosse, 2015; Nguyen et al., 2022; Perez et al., 2018; Potapczynski et al., 2023). It follows that a broadly capable learning system must be able to perform matrix operations in its internal representation, as a fundamental building block.

Not only have transformers rapidly become the most popular architecture, they are being applied to a distinctly wide array of settings. We have moved from hand-crafted feature engineering, to neural networks for particular modalities (CNNs for vision, RNNs for sequences, etc.), to *transformers for everything* (Goldblum et al., 2024). However, if transformers are to be general purpose, a pressing question arises: how good are transformers at matrix operations, a core primitive in many settings? Despite the importance of this question, there is so far relatively little work on transformers for linear algebra (e.g., Charton, 2022a;b; Yang et al., 2024).

In this paper, we first shed light on the limitations and failure modes of existing transformer approaches for matrix operations. Standard approaches involve representing a matrix as a sequence of real numbers or string tokens which is then mapped by a transformer to the output of a particular operation,

---

[*]Denotes equal contribution, names ordered by coin flip.

such as computing its eigenvalues (Charton, 2022a). These approaches have severe computational constraints, requiring a staggering $\mathcal{O}(N^4 D + N^2 D^2)$ compute and $\mathcal{O}(N^4 + N^2 D)$ memory, for an $N \times N$ matrix, and embedding dimension $D$. Moreover, when trained on standard distributions, e.g., random matrices of a fixed size with standard Gaussian entries, transformers generalize poorly to other matrix structures, even a basic identity matrix. Probing further, we show current approaches perform statistical interpolation, rather than algorithm discovery, explaining their particularly poor OOD performance.

Based on our study of these limitations, we investigate a sequence of interventions for improved scalability and performance. We note that the goal of our work is in *understanding* the potential capabilities and limitations of transformers for linear algebra, and methodological interventions are primarily in service of this understanding. First, rather than flatten the matrix, and represent each number as a string token, we embed the matrix through a learnable projection. We then pursue linear attention, which improves both scalability and the performance of matrix operations, as quadratic softmax attention is known to be poor for matrix multiplications (Arora et al., 2023; Liu et al., 2025). Together, these interventions significantly reduce compute to $\mathcal{O}(ND^2)$ and memory to $\mathcal{O}(ND+D^2)$. Next, we use a looped transformer architecture (Giannou et al., 2023; Yang et al., 2024; Saunshi et al., 2025; Geiping et al., 2025), inspired by iterative methods in numerical linear algebra (such as linear conjugate gradients, and other Krylov subspace methods (Potapczynski et al., 2023)), so that each matrix can be processed by the transformer a variable number of times, depending on the difficulty of the operation and its conditioning. Finally, we construct and train on a novel data mixture comprised of a wide range of matrix structures, covering diverse eigenspectra decays, leading to significantly better out-of-distribution generalization than the standard Gaussian random matrix training: moving from statistical interpolation towards generalizable algorithms. We illustrate the cumulative effects of these interventions in Figure 1. We name our approach the `RangeFormer` (and release code here).

We show that the `RangeFormer` profoundly relaxes the limitations of prior transformer approaches, with substantial performance improvements on diverse OOD matrices from the matrix market (Boisvert et al., 1997). Finally, for the first time, we show how transformers can be successfully used in downstream applications requiring iterative matrix operations, including Gaussian process learning, and improving the sampling distribution of randomized methods. Indeed, the ultimate test of a numerical method is whether it can support a downstream application, as part of a pipeline, especially in a setting that requires repeated matrix operations that build off of one another. In such settings, compounding errors, or systematic biases, could easily prevent convergence.

Accordingly, we summarize our key contributions here.

- **`RangeFormer` enables scalable transformers for NLA**. Past work (Charton, 2022a; Garg et al., 2023; Liu et al., 2025) has focused on matrices of size at most $50 \times 50$. Our key interventions of using a projection matrix for embedding, looping, and using linear attention enable us to train transformers for matrices up to size $1000 \times 1000$ (see Figure 6 (*Left*)). Collectively, these interventions reduce the memory complexity of the attention layer from $\mathcal{O}(N^4)$ to $\mathcal{O}(ND+D^2)$.

- **Prior models only learnt statistics of the data**. In Section 3, we demonstrate that transformers when trained on Gaussian random matrices fail to generalize on even trivial OOD examples like the Identity matrix. This is because the eigenvalues of Gaussian matrices have a predicitable distribution and thus the transformer learns to predict statistics of the training data as opposed to learning a more robust algorithm.

- **Diverse training data distribution improves OOD generalization**. To remedy the above issue, we construct a more diverse, structured training data distribution whose eigenspectra can be observed in Figure 1 *(a)*. Figure 5 (*Left*) illustrates the OOD performance gain that comes from training on this distribution.

- **`RangeFormer` can be used in downstream linear algebra applications**. For the first time, we use transformers for downstream applications of linear algebra. Specifically, we warm-start Krylov subspace methods, improve the sampling distribution in randomized SVD, and perform marginal likelihood optimizations using the RangeFormer for log-determinant calculations (Section 5).

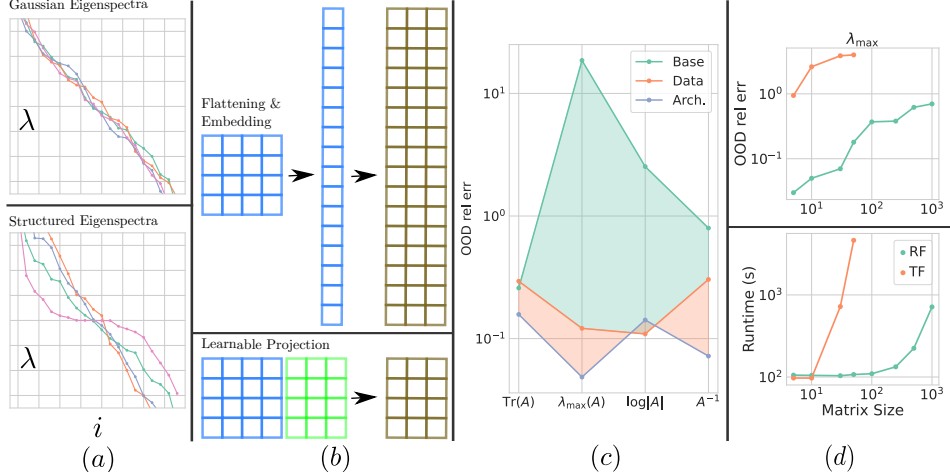

Figure 1: **How to profoundly improve the scaling and OOD performance of transformers for linear algebra.** *(a)* We train on a wide variety of structured random matrices resulting in diverse eigenspectra as opposed to only training with Gaussian random matrices. *(b)* In contrast to flattening the matrix and then projecting each scalar into a high dimensional $\boldsymbol{X} \in \mathbb{R}^{N^2 \times D}$, we embed the matrix via a learnable projection $\boldsymbol{X} \in \mathbb{R}^{N \times D}$. Together with linear attention, we achieve $\mathcal{O}(ND + D^2)$ scaling instead of $\mathcal{O}(N^4)$. *(c)* Performance improvement, over an ensemble of challenging OOD matrices (section 3), applying a set of interventions for our `RangeFormer` method, over four canonical matrix operations. The baseline (Base) is a vanilla transformer trained on Gaussian random matrices. We then train the same transformer but on our data mixture (Data) from section 4. Finally, we apply the architectural changes (Arch), such as looping, learnable projection (panel *(b)*) and linear attention. *(d)* OOD performance and runtime of `RangeFormer` (RF) against a vanilla transformer baseline (TF) from section 3. TF cannot run matrices with $N \geq 100$ on an 80 GB A100 and its performance is worse than just predicting 0. In contrast, `RangeFormer` scales reasonably in compute and performance.

## 2 TRAINING TRANSFORMERS FOR LINEAR ALGEBRA TASKS

Assume that we have data in the form of $(\boldsymbol{A}, f(\boldsymbol{A}))$ where $\boldsymbol{A}$ is a $N \times N$ matrix and $f(\boldsymbol{A})$ is a linear algebra operation applied to $\boldsymbol{A}$, say $\mathrm{Tr}(\boldsymbol{A})$, $\log |\boldsymbol{A}|$, $\lambda_{\max}(\boldsymbol{A})$, or $\boldsymbol{A}^{-1}$, and we would like to train a vanilla transformer model (TF) based on Radford et al. (2018) to approximate $f(\boldsymbol{A})$. For a given input $\boldsymbol{X} \in \mathbb{R}^{S \times D}$, where $S$ represents the sequence length and $D$ an embedding dimension, the main components of a TF are to apply a positional encoding $\boldsymbol{X} \leftarrow \mathrm{PosEnc}(\boldsymbol{X})$ and then to iterate

$$\boldsymbol{X} \leftarrow \boldsymbol{X} + \mathrm{Attn}^{(l)}(\mathrm{Norm}_A^{(l)}(\boldsymbol{X}))$$
$$\boldsymbol{X} \leftarrow \boldsymbol{X} + \mathrm{MLP}^{(l)}(\mathrm{Norm}_M^{(l)}(\boldsymbol{X})) \tag{1}$$

for $l = 1, \ldots, L$ number of blocks. Here $\mathrm{Norm}(\cdot)$ is a normalization function like layer norm, $\mathrm{Attn}(\cdot)$ is the attention mechanism (Vaswani et al., 2017) which requires $\mathcal{O}(S^2 D + S D^2)$ compute and $\mathcal{O}(S^2 + SD)$ memory. $\mathrm{MLP}(\cdot)$ is applied to the embedding dimension, thus requiring $\mathcal{O}(SD^2)$ compute and $\mathcal{O}(SD)$ memory.

We consider two baseline approaches, inspired by prior work, to train TFs for linear algebra tasks. These approaches mostly differ by how to translate the input matrix $\boldsymbol{A}$ to $\boldsymbol{X}$, and their loss functions. The first approach, following Charton (2022a) and Charton (2022b), is to train a TF with its inputs and outputs represented as strings. Here, the elements in the vocabulary $\mathcal{V}$ would be strings that represent floating-point numbers like "$\pm s_1 \ldots s_M \mathrm{E} \pm e_1 \ldots e_K$" where $s_m$ represents the matissa and $e_k$ the exponent. The size of $|\mathcal{V}|$ is approximately $\approx 2 \times 10^M \times 2 \times 10^K$ (counting redundant representations), which grows exponentially as we increase the precision of the floating-point representation by increasing $M$ and $K$. In this setup, the input matrix $\boldsymbol{A}$ gets flattened and each $A_{i,j}$ gets converted to an element of $\mathcal{V}$ and then embedded in a $D$-dimensional space to get $\boldsymbol{X} \in \mathbb{R}^{N^2 \times D}$. After $\boldsymbol{X}$ is passed into equation (1) then, assuming $f(\boldsymbol{A}) = \boldsymbol{A}^{-1}$, the last head layer creates $\boldsymbol{Y} \in \mathbb{R}^{N^2 \times |\mathcal{V}|}$ which we use to compare it to the string representation of each $A_{i,j}^{-1}$ via the cross-entropy loss. In our experiments we refer to this method as `STRFormer` (string transformer).

The second approach, `NumFormer` (numerical transformer), essentially removes the use of strings. That is, we flatten $\boldsymbol{A}$ and then embed each $A_{i,j}$ into a $D$-dimensional space through a linear layer

$\boldsymbol{W}^{(I)} \in \mathbb{R}^{1 \times D}$ to obtain $\boldsymbol{X} \in \mathbb{R}^{N^2 \times D} = \text{vec}(\boldsymbol{A})\boldsymbol{W}^{(I)}$ and, after passing $\boldsymbol{X}$ through the TF (equation 1), we obtain $\boldsymbol{Y} = \boldsymbol{X}\boldsymbol{W}^{(O)} \in \mathbb{R}^{N^2 \times 1}$ through a final linear layer $\boldsymbol{W}^{(O)} \in \mathbb{R}^{D \times 1}$. In this case we use an approximation loss between $\boldsymbol{Y}$ and $f(\boldsymbol{A})$ such as the nuclear norm $\|\boldsymbol{Y} - f(\boldsymbol{A})\|_*$ if $f(\boldsymbol{A}) = \boldsymbol{A}^{-1}$ (with $\boldsymbol{Y}$ reshaped as $N \times N$) or $|Y_{N^2} - f(\boldsymbol{A})|$ if $f(\boldsymbol{A}) \in \mathbb{R}$.

The use of flattening allows both `STRFormer` and `NumFormer` to run for any matrix size. However, following the discussion on equation 1, each TF block requires a prohibitive $\mathcal{O}(N^4 + N^2 D)$ memory. As seen in Figure 1 we cannot run matrices larger than $N > 50$ as we run out of memory.

In terms of the training data, the standard procedure in the literature is to sample random Gaussian matrices, $A_{i,j} \sim \mathcal{N}(0, 1)$. This sampling strategy is used in prior work (Charton, 2022a;b) (among other sampling distributions), and in all the works that study solving regression tasks with transformers (Garg et al., 2023; von Oswald et al., 2023; Vladymyrov et al., 2024; Fu et al., 2024; Liu et al., 2025; Yang et al., 2024). By sampling new random matrices during training we never repeat data and therefore follow a single epoch training as is done for current LLMs.

## 3    DIAGNOSING THE SHORTCOMINGS OF USING TRANSFORMERS FOR NLA

We now seek to shed light on the limitations of using transformers for numerical linear algebra tasks. We then consider several interventions in section 4 that further advance our understanding.

In this section we only show results with `NumFormer` as it consistently performs an order of magnitude better than `STRFormer`, as we will see in section 4. The main reason for the poor performance of `STRFormer` is that the string representation of the numbers sacrifices numerical precision. Thus, `NumFormer` trained with Gaussian random matrices $A_{i,j} \sim \mathcal{N}(0, \sigma)$ (with $\sigma = 1$ unless specified otherwise) is our main baseline and starting point. See Appendix D for additional experimental details and Appendix D.1 for further comparisons between `STRFormer` and `NumFormer`. While a simple but substantial improvement on `STRFormer`, and our baseline, we note `NumFormer` itself is novel and has not been proposed in prior work to the best of our knowledge.

**Are transformers learning an algorithm?**    Prior work argues how transformers appear to learn in-context regression (Garg et al., 2023) through iterative algorithms like gradient descent (von Oswald et al., 2023) or Newton's method (Fu et al., 2024). The argument in von Oswald et al. (2023) is that a simplified transformer architecture has a set of parameter combinations that represent the gradient descent algorithm and, empirically, as we traverse the blocks in equation 1, the approximations become better at a similar rate as Newton's method (Fu et al., 2024).

In our linear algebra context, if our model has learned an algorithm to solve the task, we could then pass any matrix $\boldsymbol{A}$ that is not a random Gaussian and expect a reasonable solution. Throughout our paper, we will evaluate our trained models on two sets of such OOD matrices $\mathcal{S}$ and $\mathcal{M}$. The first set $\mathcal{S}$ contains canonical structured matrices such as identity $\boldsymbol{I}$, diagonal $\boldsymbol{D}$ with $D_{i,i} \sim \mathcal{N}(0, 1)$, and Toeplitz with the bands sampled from $\mathcal{N}(0, 1)$. Even though the entries of the structured matrices in $\mathcal{S}$ are also sampled from a $\mathcal{N}(0, 1)$, their spectra are quite different, as we show in Figure 1. The second set $\mathcal{M}$ contains more than 100 non-random matrices from the *matrix market* (Boisvert et al., 1997). These matrices are gathered from challenging real-world applications like finite element approximations, structural engineering, fluid flow, power system networks and many more (see appendix B.2).

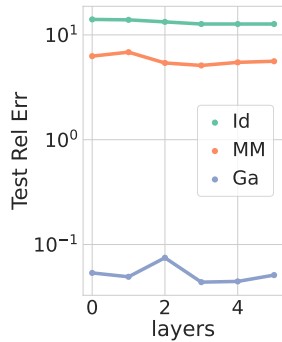

Figure 2: **The error does not decrease as we traverse through the transformer blocks.**    We plot the relative error on the inverse $\boldsymbol{A}^{-1}$ task of a `NumFormer` model for Gaussian symmetric matrices (Ga), for the Identity (Id) and the mean of the *matrix market* ensemble (MM).

For Figure 2 we train a 6 layered `NumFormer` model on $20 \times 20$ Gaussian symmetric matrices $A_{i,j} \sim \mathcal{N}(0, 1)$ to calculate the inverse $\text{NN}_\theta(\boldsymbol{A}) \approx \boldsymbol{A}^{-1}$. We compute the relative error using the nuclear norm $\text{rerr}(\boldsymbol{A}) = \|\boldsymbol{A}\,\text{NN}_\theta(\boldsymbol{A}) - \boldsymbol{I}\|_* / \|\boldsymbol{I}\|_*$, after each block of the transformer model (equation 1). We test on three different matrices: (1) the distribution of matrices that the model was trained on, Gaussian symmetric matrices, (2) the identity $\boldsymbol{A} = \boldsymbol{I}$ and (3) on all the matrices $M \in \mathcal{M}$, computing $\frac{1}{|\mathcal{M}|} \sum_{M \in \mathcal{M}} \text{rerr}(M)$. In contrast to

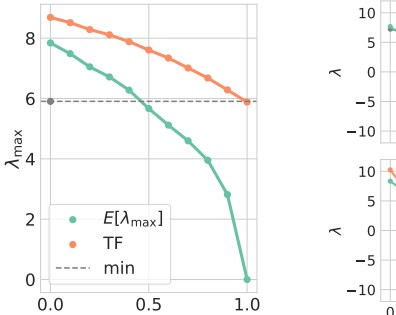 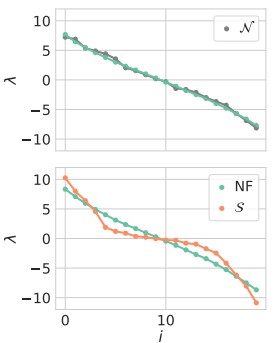 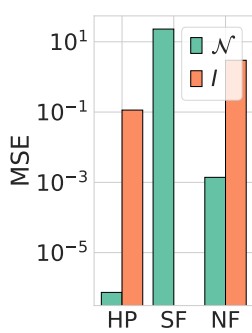

Figure 3: **Unexpected failure cases for transformer methods for NLA.** *Left:* We compute the $\lambda_{\max}$ for distinct symmetric Gaussian matrices $A_{i,j} \sim \mathcal{N}(0,1)$ as we mask their inputs with different proportions of non-zero values (green), compared to the output of a `NumFormer` model trained on the same symmetric Gaussian matrices (orange). As we increase the proportion of zeros in the input, the `NumFormer` models goes from predicting $E[\lambda_{\max}]$ without masking to $\min \lambda_{\max}$. *Middle:* In the top panel, we compare the full eigenspectrum of a symmetric Gaussian matrix $A_{i,j} \sim \mathcal{N}(0,1)$ (grey) to the eigenspectrum prediction for the same matrix for a `NumFormer` model trained on the same distribution of symmetric Gaussian matrices (green). The `NumFormer` learns a linear decay in the spectrum that closely matches the training data. In the bottom panel, we compare the eigenspectrum prediction of a `NumFormer` but now for a symmetric Toepliz matrix with Gaussian bands $A_{i,j} \sim \mathcal{N}(0,1)$ (orange). The `NumFormer` erroneously predicts the same linear decay for the Toepliz input. *Right:* We display the MSE loss of 3 different methods (Liu et al., 2025; Charton, 2022a;b) for training transformers to solve least-squares problems when we pass an in-distribution Gaussian matrix (green) versus passing the identity matrix (orange). There is significant performance degradation when evaluating on canonical matrices like the identity and `STRFormer` failed to decode for $I$. See section 3 for full details.

von Oswald et al. (2023) and Fu et al. (2024), we do not observe a monotonically decreasing error as we traverse the different layers in the transformer. This finding suggests that the `NumFormer` model is at least not learning an iterative algorithm to solve the task. Moreover, we see the `NumFormer` has high error for the OOD matrix market, and even worse for the trivial identity matrix! We revisit this observation shortly, in validating with canonical matrices (below).

**Transformers as Statistical Machines** From random matrix theory, we know that a symmetric Gaussian matrix $A_{i,j} \sim \mathcal{N}(0, \sigma^2)$ with $A_{i,j} = A_{j,i}$ has the bulk of its eigenvalues $\lambda_i(A)$ in the interval $(-2\sigma\sqrt{N}, 2\sigma\sqrt{N})$, and $\mathbb{E}[\lambda_{\max}(A)]/\sqrt{N} \to 2\sigma$ as $N \to \infty$ (Tao, 2012). In Figure 3 *(left)* we train a `NumFormer` model on symmetric $20 \times 20$ $A_{i,j} \sim \mathcal{N}(0,1)$ matrices to solve $\lambda_{\max}$. We then probe its predictions on a set of masked inputs $M \odot A$ where $M_{i,j} \in \{0,1\}$ and vary the overall proportion of nonzero entries. That is, we go from the zero matrix $0 = 0 \odot A$ (masking proportion = 1) to $A = 1 \odot A$ (masking proportion = 0). Note how the `NumFormer` model assigns to the zero matrix an estimate of the min $\lambda_{\max}(A)$ most likely obtained through training. In other words, $\text{NN}_\theta(0) \approx \min(\lambda_{\max}(A_b))_{b=1}^B$ (grey dot and line), which is much larger than $\lambda_{\max}(0) = 0$. We compute our estimate with $B = 10K$.

In Figure 3 *(middle)* we train a `NumFormer` model on symmetric $20 \times 20$ $A_{i,j} \sim \mathcal{N}(0,1)$ matrices to predict the whole eigenspectrum $\lambda(A) \in \mathbb{R}^{20}$. In the top panel we have the prediction of the model $\text{NN}_\theta(A_\mathcal{N})$ and the actual spectrum $\lambda(A_\mathcal{N})$ for $A_\mathcal{N} \sim \mathcal{N}(0, I)$ and $A_\mathcal{N}^\top = A_\mathcal{N}$. On the bottom panel we have the prediction of the model and the actual spectrum but for a Toeplitz structured matrix $A_\mathcal{S} \in \mathcal{S}$. The `NumFormer` learned through training that a linear decay is a close approximation to the $\lambda(A_\mathcal{N})$ and it erroneously predicts a linear decay also for $A_\mathcal{S}$.

Overall, we found these results alarming as the transformer models appear to be learning statistical properties of the random matrices that they were trained on and not an algorithm to solve the linear algebra operations. Similar results can be obtained when $A_{i,j}$ is sampled from a Laplace or uniform distribution (see Appendix D.1).

**Validating with Canonical Matrices** It is common to sanity check an NLA algorithm on matrices like the identity $A = I$, zero $A = 0$, or diagonal $A = \text{diag}(d)$, as these are considered trivial matrices with closed-form solutions. In Figure 3 *(left)*, we show the discrepancy between the test loss achieved in-distribution versus the loss obtained by passing $A = I$ as the input

for distinct transformer models for least-squares problems $\|\boldsymbol{Ax} - \boldsymbol{b}\|_2$. Namely, `NumFormer`, `STRFormer` (Charton, 2022a;b), and the high-precision solver (`HPS`) in Liu et al. (2025).

We train our models following the setup in Liu et al. (2025) which is also almost identical to Charton (2022a) and Vladymyrov et al. (2024). That is, $\boldsymbol{A} \in \mathbb{R}^{20 \times 5}$, $\boldsymbol{A}_{i,j} \sim \mathcal{N}(0,1)$, $\boldsymbol{b} \in \mathbb{R}^5$, and $b_i \sim \mathcal{N}(0,1)$ where we cap the condition number to $\kappa(\boldsymbol{A}) = 5$. In Figure 3 *(right)* we see how all the methods struggle to produce reasonable outputs for $\boldsymbol{A} = \boldsymbol{I}$. On the one hand, $\boldsymbol{I}$ is OOD since matrices close to $\boldsymbol{I}$ have low probability density under $\boldsymbol{A}_{i,j} \sim \mathcal{N}(0,1)$. Yet, it is surprising if we believe that the transformer models are learning a reasonable algorithm, as $\boldsymbol{I}$ should be a trivial example.

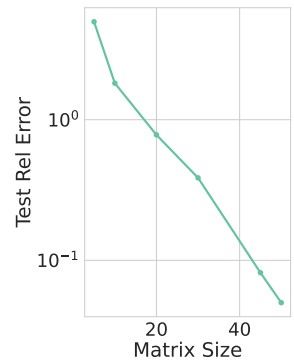

Figure 4: **The approximation error exponentially increases for smaller matrices.**

**On Different Matrix Sizes** A clear benefit of transformers is that they can handle different sequence lengths, here different matrix sizes, without any additional intervention. When training a LLM such as a GPT-2 (Radford et al., 2018), we can create input batches of the same context length by collecting sequences of different lengths and denoting the length difference using the `<EOS>` token. When training transformers for linear algebra problems, it is not clear how to accommodate for different matrix sizes in the same batch, so the transformers models in the literature are trained with a fixed size. In Figure 4 we train a `NumFormer` model on $50 \times 50$ Gaussian symmetric matrices $A_{i,j} \sim \mathcal{N}(0,1)$ on the Tr($\boldsymbol{A}$) task. We measure the relative error when passing as input Gaussian symmetric matrices but of sizes $5 \times 5$, $10 \times 10$, $20 \times 20$, $30 \times 30$, $45 \times 45$ and $50 \times 50$. Even a reduction from $N = 50$ to $N = 45$ worsens the performance almost 10 times.

## 4 THE RANGEFORMER METHOD

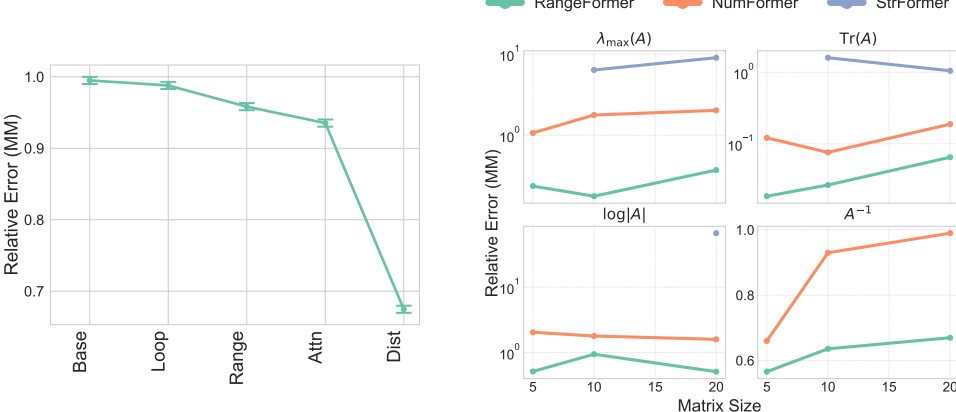

Figure 5: **Our data and architectural interventions significantly improve performance.** *(Left):* Intervention breakdown and its effect on improving the test relative error on OOD *matrix market* data for the $\boldsymbol{A}^{-1}$ task for $20 \times 20$ matrices. Our baseline is the `NumFormer` model trained on Gaussian random matrices, and the interventions are: (1) adding an iterative bias (*Loop*), (2) embedding through the range of the operator (*Range*), (3) incorporating subquadratic attention mechanisms (*Attn*), (4) using our data distribution (*Dist*). *(Right):* Performance of `RangeFormer` against `STRFormer` (Charton, 2022a) and `NumFormer` on $\lambda_{\max}(\boldsymbol{A})$, Tr($\boldsymbol{A}$), $\log |\boldsymbol{A}|$ and $\boldsymbol{A}^{-1}$ across *matrix market* data for sizes $5 \times 5$, $10 \times 10$ and $20 \times 20$. We expand upon the different model configurations and loss functions in appendix D.7. The missing numbers for `STRFormer` are due to decoding errors.

We now construct, step-by-step, our `RangeFormer` architecture, and introduce our synthetic data procedure. We will study the effects of each intervention on classical linear algebra problems such as max eigenvalue $\lambda_{\max}(\boldsymbol{A})$, trace Tr($\boldsymbol{A}$), log determinant $\log(|\boldsymbol{A}|)$ and inverse $\boldsymbol{A}^{-1}$ (additional details in Appendix D.2).

Figure 5 *(left)* shows the cumulative and substantial performance increase from all the interventions, previously summarized in Figure 1, that transform `NumFormer` to `RangeFormer`. In Figure 5 *(right)* we compare the performance of `RangeFormer` across different matrix sizes and linear algebra operations. For each linear algebra operation, we train each model using our batching technique (discussed below) that enables the models to perform for different matrix sizes and avoids the lack of size generalization discussed in Figure 4. Overall, `RangeFormer` is an order of magnitude better when testing our OOD *matrix market* ensemble. For `STRFormer`, we use the code and checkpoints provided by Charton (2022a;b). Interestingly, the checkpoints perform well when ran on in-distribution random matrices but they usual fail to decode a correct string when ran on the OOD *matrix market* data. We now explain each intervention in detail.

**Positional encodings**    Though not a particular intervention in Figure 5, introducing positional encodings is vital for our application as the results of linear operations like trace, solves, or max eigenvalue are not permutation invariant. For example, permuting the identity changes the value of the trace. In appendix D.3 we compare the performance of the different encoding methods.

**Looping**    To introduce an algorithmic bias into our transformer model, and avoid the situation depicted in Figure 2, we tie the transformer weights in equation 1 to obtain a *looped transformer* (Saunshi et al., 2025; Giannou et al., 2023; Yang et al., 2024; Geiping et al., 2025). Via looping, now the model is *forced* to use the recurrences to improve the solutions, similar to how most iterative linear algebra routines work, such as linear conjugate gradients, or stochastic Lanczos quadrature (Saad, 2003). In Figure 6 *(left)*, we compare the quality of the approximation of the transformer model as we traverse the layers in equation 1 on a OOD Toepliz matrix for the $A^{-1}$ task. The `NumFormer` model does not improve its approximation as we traverse the layers. In contrast, our `RangeFormer` model monotonically decreases the error as we traverse the layers, more sharply in the beginning and flattening at the end. Additional experimental details are in Appendix A.2.

**Range embedding**    This intervention circumvents the $\mathcal{O}(N^4 + N^2 D)$ scaling of the model by not flattening the matrix. Rather, we learn a projection matrix $\Gamma \in \mathbb{R}^{N \times D}$ that allows us to work directly with the range of the operator as $X = A\Gamma \in \mathbb{R}^{N \times D}$ by now considering the rows of $X$ as the sequence dimension and the columns as the embedding, scaling as $\mathcal{O}(N^2 + ND)$. To underscore the relevance of this modification, we name our transformer models as `RangeFormer`. Figure 5 *(left)* shows a minor performance improvement in substituting to this approach despite a reduction in total parameter count. See Appendix D for additional details on the architecture.

**Attention alternatives**    Giannou et al. (2023) and Liu et al. (2025) argue that the scale normalization and the nonlinear functional form of the standard softmax attention introduces an approximation distortion that is not favorable to linear algebra primitives like matrix-multiplies. Thus, we use linear attention alternatives like Taylor attention (Arora et al., 2023) or `BaseConv` (Liu et al., 2025) that allow us to improve upon standard attention and, most notably, achieve additional scaling improvements of $\mathcal{O}(ND + D^2)$ by avoiding the construction of the full attention matrix. Figure 1 *(c)* shows the performance improvement captured by this architectural intervention (Arch) (among others). Moreover, Figure 1 *(d)* shows the staggering runtime and memory benefits achieved through this intervention. We can now train transformers on matrices of size $N = 1K$ in less time than prior methods on $N = 50$ as seen in Figure 1 (d). Additional large scale experiments in Appendix D.1.

**Training data distribution**    As discussed in section 3, a notable limitation of previous methods is to solely train on Gaussian random matrices (Liu et al., 2025; Charton, 2022a;b; Dutta and Sra, 2025; Vladymyrov et al., 2024) as transformer models would then only learn statistical properties about the data. As seen in Appendix B.3, Figure 8, different Gaussian matrices almost have an identical eigenspectrum, making them essentially the same linear operator. We take a different approach and construct a data mixture consisting of two main elements: structured matrices and matrices with different eigenspectrum decays. To generate structured matrices we use the continuous Einsum parameterization from Potapczynski et al. (2024) (discussed in Appendix B) which allows us to randomly sample structures like Kronecker, Low-rank, Tensor Train, Block Tensor Train and Monarch, and many more. Interestingly, our way of sampling from the Einsum parameterization prevents us from sampling matrices like $I$, $0$, diagonal, or Toeplitz (Appendix B). Furthermore, in Appendix B.3, Figure 9, we see how the different structures lead to diverse eigenspectra, even when the elements of these matrices are still sampled from $\mathcal{N}(0, 1)$. Finally, as discussed in Appendix B.3, we use a series of functional forms to create a wide variety of eigenspectra $\Lambda$ as seen in Figure 10. Once we have $\Lambda$, then we sample a random basis $Q$ and create $A = Q^\intercal \Lambda Q$. Figure 6 *(Right)*

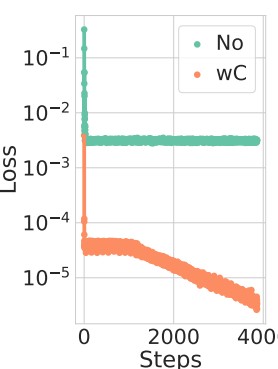 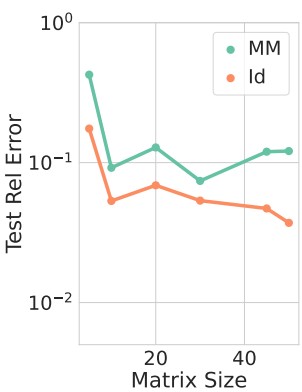 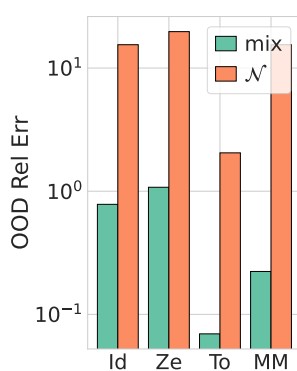

Figure 6: **Different benefits of training with our methodology.** *(Left):* Curriculum learning enables our models to learn problems at larger scales. Train loss of `RangeFormer` for the linear solves task in (Liu et al., 2025). *(Middle):* We compare the performance on the same two OOD cases, of our `RangeFormer` model across different matrix sizes. The `RangeFormer` model was trained using our batching method on $50 \times 50$, $30 \times 30$ and $10 \times 10$ matrices but can perform relatively well for other cases like $5 \times 5$, $20 \times 20$ and $45 \times 45$. *(Right):* Incorporating a diversity of structures, as well as matrices with a wide variety of eigenspectra decay, in the training distribution highly improves OOD performance. We show the relative error on different OOD matrices for a `RangeFormer` model trained on Gaussian random matrices and one trained on our data mixture (mix) for $\lambda_{\max}$. The matrices are the Identity (Id), zeros (Ze), Toeplitz (To) and the *matrix market* ensemble (averaged).

shows the benefits of transitioning from training a `RangeFormer` model on Gaussian random data, to one trained on our data mixture for the inverse $A^{-1}$ task. Also Figure 1 *(c)* displays the benefits of training with this data mixture on other linear algebra operations.

**Curriculum learning and varying sequence length**     At the $N = 1K$ scale, particularly for solving linear systems, we could not get our `RangeFormer` models to converge unless we first got a model checkpoint that solves $N = 100$ and then use that checkpoint as a starting point to solve $N = 1K$. In Figure 6 *(Left)* we see how only when doing the curriculum would the loss start to decrease. To avoid the situation depicted in Figure 4, we train our models with a diverse set of matrix sizes $\{N_1, \ldots, N_R\}$. Figure 6 *(Middle):* shows how our model can stabilize its OOD performance across different sizes, even ones that were not seen during training.

## 5   DOWNSTREAM APPLICATIONS

Ultimately, matrix operations are rarely an end in and of themselves but instead are used as part of a downstream application. For the first time, we replace linear algebra routines with transformers to compute the matrix operations in downstream applications.

### 5.1   ACCELERATING THE CONVERGENCE OF KRYLOV SUBSPACE METHODS

Krylov subspace methods are classical iterative methods for computing matrix operations, such as conjugate gradients (CG) for solving linear systems. As discussed in appendix A.1, any Krylov subspace algorithm requires an initial vector to start forming the basis $\mathcal{K}^{(t)}(x_0, A)$. We now use our trained `RangeFormer` models $\mathrm{NN}_\theta(\cdot)$ to provide $x_0 = \mathrm{NN}_\theta(A)$. The intuition is that since $\mathrm{NN}_\theta$ provides a good approximation to the solution, then the overall effect is that the Krylov subspace algorithm will converge to the solution faster when compared to simply picking a standard random or zero initialization for $x_0$. In Figure 7 *(Right)* we show how CG converges much rapidly on the *bcsstk02* matrix from the *matrix market*. We provide additional results for other algorithms and other matrices in appendix D.4. Since matrix operations are a fundamental primitive for learning tasks, our goal in this paper is to expose limitations of transformers for linear algebra, and show how they can be improved, rather than to argue that they are preferable to classical solvers. However, this experiment demonstrates that there is indeed promise for transformers to be combined with classical solvers for better performance than these solvers could achieve alone.

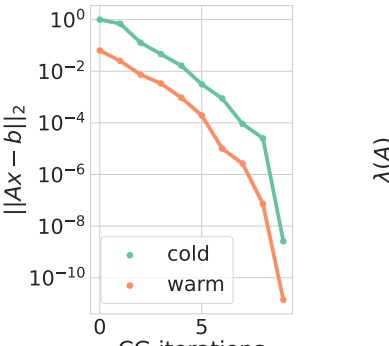 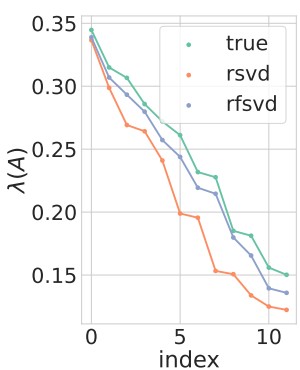 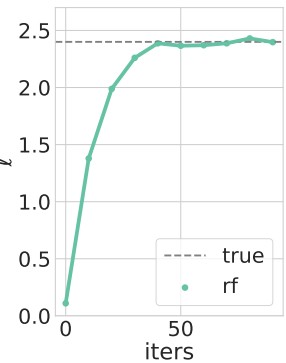

Figure 7: **Performance of `RangeFormer` on downstream applications.** *(Left):* `CG` convergence with and without `RangeFormer` warm starts on *bcsstk02* from the *matrix market. (Middle):* Spectrum approximation for `RSVD` sampling $\boldsymbol{\Omega} \sim \mathcal{N}(\mathbf{0}, \boldsymbol{I})$ and sampling $\boldsymbol{\Omega} \sim \mathrm{NN}_\theta(\boldsymbol{A})$. *(Right):* The `RangeFormer` linear algebra primitives allow us to recover the length scale parameter of an RBF kernel for data generated from a Gaussian process with an RBF kernel.

## 5.2 IMPROVING THE SAMPLING DISTRIBUTION FOR RNLA

All randomized linear algebra methods rely on sampling some random embedding $\boldsymbol{\Omega}$, usually Gaussian $\boldsymbol{\Omega} \sim \mathcal{N}(\mathbf{0}, \boldsymbol{I})$. In this application, we train `RangeFormer` models that define $p(\boldsymbol{\Omega}) = \mathrm{NN}_\theta(\boldsymbol{A})$ such that our samples $\boldsymbol{\Omega} \sim p(\boldsymbol{\Omega})$ improve the performance of the randomized algorithm. In other words, $\mathrm{NN}_\theta$ will take into consideration the characteristics and structure of $\boldsymbol{A}$ and bias $p(\boldsymbol{\Omega})$ to provide a better sampling distribution. Since `RangeFormer` uses looping, the process of sampling $\boldsymbol{\Omega}$ is similar to diffusion, where we start with $\boldsymbol{\Omega_0} \sim \mathcal{N}(\mathbf{0}, \boldsymbol{I})$ and progressively pass this noise through the layers of $\mathrm{NN}_\theta$ to get $\boldsymbol{\Omega} \sim \mathrm{NN}_\theta(\boldsymbol{A})$. In Figure 7 *(Middle)*, we compare the performance of randomized SVD using `RangeFormer`, namely `RFSVD`, and the usual algorithm `RSVD` that uses $\boldsymbol{\Omega} \sim \mathcal{N}(\mathbf{0}, \boldsymbol{I})$. We use a matrix $\boldsymbol{A} \in \mathbb{R}^{100 \times 100}$ stemming from a Gaussian Process RBF kernel on a uni-variate grid that has a fast decaying spectrum. As seen in the figure, `RFSVD` (with $D = 16$) is able to more closely resemble the spectrum of $\boldsymbol{A}$ when compared to a `RSVR` (rank 16) approximation (appendix D.5 has additional details). Moreover, Figure 15 *(Right)* also shows contour plots of the learned distribution $p(\boldsymbol{\Omega})$ which are quite distinct from the contours of $\mathcal{N}(\mathbf{0}, \boldsymbol{I})$ Figure 15 *(Left)*.

## 5.3 GAUSSIAN PROCESSES

In our last application, we use `RangeFormer` to train Gaussian process (GPs) kernel hyperparameters. This is a particularly challenging stress-test, as the kernel learning pipeline requires a sequence of matrix operations that build off one another, where compounding errors and biases could preclude convergence.

GPs are flexible distributions over functions, with properties controlled by a kernel (Rasmussen and Williams, 2006). Given some data $(x_i, y_i)_{i=1}^N$ and a kernel function $k_\phi(x, x')$ parametrized by $\phi$, training a GP requires constructing $\boldsymbol{K}_\phi[i, j] = k_\phi(x_i, x_j)$ and then minimizing the negative log-likelihood $\mathcal{L}(\phi) = \log|\boldsymbol{K}_\phi| + \boldsymbol{y}^\mathsf{T} \boldsymbol{K}_\phi^{-1} \boldsymbol{y}$ which involves the computation of log-determinants and solves (Rasmussen and Williams, 2006). In our case, we use `RangeFormer` for these operations in order to learn $\phi$. In Figure 7 *(Right)* we perform a sanity check showing that our `RangeFormer` pipeline recovers the true length scale $\ell$ when trained on data generated from a GP. Finally, we fit a RBF kernel on data generated as $y_i = \sin(2\pi x_i) + \epsilon_i$, for $i = 1, \ldots, 50$ with $\epsilon_i \sim \mathcal{N}(0, 0.04)$, and $x_i \sim \mathcal{U}[0, 1]$, using our `RangeFormer` pipeline and using `Cholesky`. Both strategies give almost identical results on the hyperparameter values learned, their trajectories, and the test RMSEs: 0.87996 (`Cholesky`) and 0.87978 (`RangeFormer`). We provide additional experimental details in appendix D.6.

## 6 DISCUSSION

We have found that current transformer approaches for matrix operations profoundly fail on even trivial out-of-distribution matrices, such as the identity matrix. We shed light on these failure modes, showing that current approaches perform statistical interpolation on in-distribution matrices, rather discovering more generalizable algorithms. We also find that current approaches are computationally intractable, beyond even tiny $30 \times 30$ matrices.

Through a sequence of interventions, we were able to profoundly relax these limitations. In particular, we found that introducing a learnable projection improves scaling by several orders of magnitude, and new richer training distributions improve performance by several orders of magnitude, especially in OOD settings. These interventions are not merely making transformers better at numerical linear algebra, but moving from a system that was essentially broken, to one that can start to perform competently on challenging real-world matrices, and shifting closer to algorithm discovery.

This research area is in its early stages. Our goal is not to compete with purpose-built classical algorithms, just like a transformer could not be expected to compete with a calculator for arithmetic. At the same time, we also provide the first results using transformers successfully for matrix operations in downstream applications, which suggests that transformers could become practical in this setting. For example, GP marginal likelihood optimization is particularly challenging, since it is an iterative pipeline that would quickly become unstable under compounding errors. Moreover, our randomized linear algebra application also shows how transformers can play a complementary role to classical methods, learning how to improve randomized SVD. And our warm-start CG application additionally shows how transformers can naturally be combined with classical methods.

We hope that our results will inspire research into transformers for matrix operations. Presently, this research area is in its infancy, relative to decades of work on purpose-built classical solvers, representing millions of human hours of effort. But with a sustained program, over years, we will see both more general purpose architectures, and transformers that become practically compelling for matrix operations. Indeed, after years of work on neural graph methods, which initially were outperformed by classical approaches, there were eventually significant breakthroughs.

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

## APPENDIX OUTLINE

1. Appendix A provides additional background on Linear Algebra problems and methods.
2. Appendix B describes our data construction procedure, evaluation metrics, and provides an analysis of the the eigenspectra of different data distributions.
3. Appendix C describes the proposed `RangeFormer` architecture in detail.
4. Appendix D provides additional experimental details.
5. Appendix E formally analyzes the runtime for `NumFormer` and `RangeFormer`.

## A  BACKGROUND

In this section we briefly introduce and describe the techniques, methods and problems that will be used throughout the paper.

## A.1 LINEAR ALGEBRA

**Canonical Linear Algebra problems**    Assume that we have $\boldsymbol{A} \in \mathbb{R}^{N \times N}$. Arguably the most important linear algebra operations, besides matrix-vector-multiplies (MVM) $\boldsymbol{A}\boldsymbol{x}$, are: (1) linear solves $\boldsymbol{x} = \boldsymbol{A}^{-1}\boldsymbol{b}$, (2) eigen decompositions $\boldsymbol{A} = \boldsymbol{Q}\boldsymbol{\Lambda}\boldsymbol{Q}^{\mathsf{T}}$ (assuming $\boldsymbol{A}$ is symmetric), (3) applying a function $f$ to $\boldsymbol{A}$, as $f(\boldsymbol{A}) = \boldsymbol{Q}f(\boldsymbol{\Lambda})\boldsymbol{Q}^{\mathsf{T}}$ or (4) applying a matrix to scalar function like trace $\mathrm{Tr}(\boldsymbol{A})$, determinant $|\boldsymbol{A}|$, or condition number $\kappa(\boldsymbol{A})$. The previous list is far from exhaustive but it encompasses the majority of the operations that are common place in ML. For example, linear solves and log-determinant computations are part of Gaussian Process (GP) training (Rasmussen and Williams, 2006) while eigen-decompositions and matrix functions $f(\boldsymbol{A})$ are part of approximate second-order optimizers like Shampoo (Anil et al., 2020). In section 4 we will train transformers to tackle all of the aforementioned operations.

**Linear Algebra methods**    There is a vast set of algorithms to solve all the linear algebra problems from above (Trefethen and Bau, 1997; Golub and Loan, 2018). In this paper, we will focus on two broad families of methods that are the most commonly used in practice: (1) Krylov subspace methods (Saad, 2003; 2011) and (2) randomized methods (Martinsson and Tropp, 2020).

The main idea of Krylov subspace methods is to iteratively apply MVMs to construct a subspace that can be used to solve a target problem. Namely, given an initial vector $\boldsymbol{x}_0$, and a number of iterations $t = 1, \ldots, N$, we will implicitly construct and utilize the Krylov subspace $\mathcal{K}^{(t)}(\boldsymbol{x}_0, \boldsymbol{A}) = \mathrm{span}\{\boldsymbol{x}_0, \boldsymbol{A}\boldsymbol{x}_0, \ldots, \boldsymbol{A}^{t-1}\boldsymbol{x}_0\}$. Based on the previous equation, the cost of these methods can thus be upper-bounded by $\mathcal{O}\left(TN^2\right)$ by assuming the worst case MVM cost for $\boldsymbol{A}$ and where $T$ is the number of iterations. To give some examples, some well-known Krylov subspace methods for linear solves are `CG`, `MINRES`, and `GMRES` (Saad, 2003) and for eigenvalue problems we have the `Power Iteration`, `Lanczos`, `LOBPCG`, and `Arnoldi` (Saad, 2011). In section 5.1 we will train transformers to suggest an initial $\boldsymbol{x}_0$ that will accelerate the convergence of these methods.

The key insight of randomized methods is to reduce the problem size by multiplying $\boldsymbol{A}$ with a random matrix $\boldsymbol{\Omega} \in \mathbb{R}^{N \times \ell}$ where $\ell \ll N$ and then apply distinct deterministic methods to $\boldsymbol{Y} = \boldsymbol{A}\boldsymbol{\Omega} \in \mathbb{R}^{N \times \ell}$ to solve a target problem. A famous example is `RSVD` (Martinsson and Tropp, 2020), where we extract an orthonormal basis $\boldsymbol{Q} = \mathrm{QR}(\boldsymbol{Y})$, then construct $\boldsymbol{C} = \boldsymbol{Q}^{\mathsf{T}}\boldsymbol{A} \in \mathbb{R}^{\ell \times N}$, get $[\boldsymbol{U}, \boldsymbol{\Sigma}, \boldsymbol{V}] = \mathrm{SVD}(\boldsymbol{C})$ and thus provide an approximation of the form $\boldsymbol{A} \approx (\boldsymbol{Q}\boldsymbol{U})\boldsymbol{\Sigma}\boldsymbol{V}^{\mathsf{T}}$ where $\boldsymbol{U} \in \mathbb{R}^{\ell \times \ell}$, $\boldsymbol{V} \in \mathbb{R}^{N \times \ell}$ are orthonormal matrices, and $\boldsymbol{\Sigma} \in \mathbb{R}^{\ell \times \ell}$ is a diagonal matrix. With `RSVD`, we avoid the $\mathcal{O}(N^3)$ cost of applying `SVD` to $\boldsymbol{A}$ but rather incur a $\mathcal{O}(\ell^2 N)$ cost of the `SVD` to $\boldsymbol{C}$. Most commonly, $\boldsymbol{\Omega}$ is sampled $\boldsymbol{\Omega} \sim \mathcal{N}(\boldsymbol{0}, \boldsymbol{I})$ but there are also other more efficient alternatives such as sparse sign matrices or subsampled trigonometric transforms (Martinsson and Tropp, 2020). However, the overall decision of the sampling distribution is either speed or convenience and, in section, 5.2 we explore other learnt distributions that our transformer models uncovered to improve performance.

**Matrix structures**    By matrix structure we refer to various patterns and relations between the entries of $\boldsymbol{A}$ that ultimately lead to more efficient MVMs and more efficient algorithms to solve distinct linear algebra problems. A trivial example would be when $\boldsymbol{A}$ is diagonal, that is $A_{i,j} = 0$ if $i \neq j$. In this case, a MVM simply amounts to $\boldsymbol{A}\boldsymbol{x} = \mathrm{diag}(\boldsymbol{A}) \odot \boldsymbol{x}$, a $\mathcal{O}(N)$ operation not $\mathcal{O}(N^2)$. Additionally, a linear solve would also be $\mathcal{O}(N)$, since $\boldsymbol{x} = \mathrm{diag}(\boldsymbol{A})^{-1} \odot \boldsymbol{b}$. A popular example is when $\boldsymbol{A}$ is Toeplitz. In this case, the `FFT` is used to efficiently compute a MVM in $\mathcal{O}(N \log N)$. In section 4 we will make extensive use of different matrix structures to pre-train the transformers. Specifically, we will use the Einsum parameterization of matrix structures from Potapczynski et al. (2024). The Einsum parameterization is able to capture many well-known structures such as low rank, Kronecker, Tensor Train (Oseledets, 2011), Monarch (Dao et al., 2022), BTT (Qiu et al., 2024) and more by representing the structures as tensor contractions between two factors. All that is required in the Einsum parameterization is to specify a real valued vector $\boldsymbol{\xi} \in [0, 1]^7$ and the values on the entries $\xi_i$ dictate the structure of the matrix.

## A.2 TRANSFORMERS

**Architecture**    Our architecture is inspired and resembles Radford et al. (2018). That is, for a given input $\boldsymbol{X} \in \mathbb{R}^{S \times D}$, where $S$ represent the sequence length and $D$ an embedding dimension, we will

first apply a positional encoding $\boldsymbol{X} \leftarrow \text{PosEnc}(\boldsymbol{X})$ and then iterate

$$\boldsymbol{X} \leftarrow \boldsymbol{X} + \text{Attn}^{(l)}(\text{Norm}_A(\boldsymbol{X}))$$

$$\boldsymbol{X} \leftarrow \boldsymbol{X} + \text{MLP}^{(l)}(\text{Norm}_M(\boldsymbol{X}))$$

for $l = 1, \ldots, L$ number of blocks. Here $\text{Norm}(\cdot)$ is some sort of normalization function like layer norm, $\text{Attn}(\cdot)$ refers to the attention mechanism that is detailed in the next subsection and $\text{MLP}(\cdot)$ is an MLP applied to the embedding dimension, thus requiring $\mathcal{O}(SD^2)$ operations.

We depart from Radford et al. (2018) in several ways. First, our transformer is not a text to text model. At a high level, representing numbers as strings as Charton (2022a), has two clear drawbacks: (1) the vocabulary grows exponentially in the precision and (2) it adds redundant, lossy transformations like representing numbers as strings to then embed them as numbers again. In terms of the vocabulary, if we have a string: "$\pm s_1 \ldots s_M \text{E} \pm e_1 \ldots e_K$", then the size of the vocabulary would be approximately $|V| \approx 2 \times 10^M \times 2 \times 10^K$ (counting redundant representations). Second, our transformer model does not apply a causal mask as the sequence length is related to the input matrix where we have complete access. Third, our training loss is not next token prediction but rather approximation error. Arguably the deviations from Radford et al. (2018) make our models completely different in nature. However, we are not trying to train a LLMs to do linear algebra, rather we are training neural networks to solve linear algebra tasks and a transformer architecture is a natural choice as it can gracefully handle different sequence lengths (matrix sizes).

**Attention mechanisms** Following the classical attention mechanism from Vaswani et al. (2017), we construct $\boldsymbol{Q}^{(h)} = \boldsymbol{X}\boldsymbol{W}^{\boldsymbol{Q}(h)}$, $\boldsymbol{K}^{(h)} = \boldsymbol{X}\boldsymbol{W}^{\boldsymbol{K}(h)}$ and $\boldsymbol{V}^{(h)} = \boldsymbol{X}\boldsymbol{W}^{\boldsymbol{V}(h)}$ with $\boldsymbol{W}^{\boldsymbol{Q}(h)}, \boldsymbol{W}^{\boldsymbol{K}(h)}, \boldsymbol{W}^{\boldsymbol{V}(h)} \in \mathbb{R}^{D \times d}$ where $d$ represents the head dimension with $Hd = D$ and $h = 1, \ldots, H$ indexes the head number. For each head, we then output $\boldsymbol{Y}^{(h)} = \text{softmax}(\boldsymbol{Q}^{(h)}\boldsymbol{K}^{(h)\mathsf{T}})\boldsymbol{V}^{(h)}$ and finally concatenate the output as $\boldsymbol{Y} = \text{concat}(\boldsymbol{Y}^{(1)}, \ldots, \boldsymbol{Y}^{(H)}) \in \mathbb{R}^{S \times D}$. This attention mechanism requires $\mathcal{O}(S^2 D)$ operations for computing $\boldsymbol{Q}^{(h)}\boldsymbol{K}^{(h)\mathsf{T}}$ for each $h$.

There have been several efforts to reduce the quadratic requirement of the standard attention mechanism, most notably the use of linear attention (Wang et al., 2020) where the goal is to circumvent the construction of $\text{softmax}(\boldsymbol{Q}^{(h)}\boldsymbol{K}^{(h)\mathsf{T}})$ by instead applying a feature map $\psi(\cdot)$ such that $\boldsymbol{Y}^{(h)} = \psi(\boldsymbol{Q}^{(h)})(\psi(\boldsymbol{K}^{(h)})^{\mathsf{T}}\boldsymbol{V})$ and therefore only require $\mathcal{O}(SD^2)$ operations. Two popular feature maps are $\psi(x) = 1 + \text{elu}(x)$ and $\psi(x) = 1 + x + \frac{1}{2}x^2$ (Arora et al., 2023).

**Looped transformers** The idea of *looped transformers* (Giannou et al., 2023; Yang et al., 2024; Saunshi et al., 2025; Geiping et al., 2025) is to incorporate an algorithmic inductive bias into the transformer architecture by sharing weights across blocks. Namely, $\text{Attn}^{(l)}(\cdot) = \text{Attn}(\cdot)$ and $\text{MLP}^{(l)}(\cdot) = \text{MLP}(\cdot)$ for all $l = 1, \ldots, L$. Interestingly, this inductive bias allow us to increase the amount of computation (by extending the number of loops) and hopefully improve predictions.

In our models we add the regularization techniques from Geiping et al. (2025) of injecting input noise at each iteration as well as sampling randomly the number of blocks to run at each iteration during training. We used looped transformers as they are the model class that gave us better empirical results but also because we want our transformer models to learn iterative algorithms.

## B DATASETS

### B.1 TRAINING DATA

Previous results on transformers for NLA have used random matrix families to train their models. However, random matrices exhibit certain statistical properties which may make tasks like $\lambda_{\max}(\boldsymbol{A})$ predictable from those statistics rather than required an algorithm to be learned. To avoid this pitfall, we train transformers on a distribution of structured matrices with a rich variety of eigenspectra, making this data less predictable from statistics only.

- **Base**. Our baseline training dataset simply consists of square Gaussian matrices, $\boldsymbol{A}_{ij} \sim \mathcal{N}(0, \sigma^2)$. Usually, we would symmetrize the input $\boldsymbol{A}_{i,j} = \boldsymbol{A}_{j,i}$ to have well-defined data for the problem.

- **Einsums**. The Einsum parameterization of matrices described in Potapczynski et al. (2024) captures structured matrices such as Kronecker, Low-rank, Tensor-train and many more. To sample a matrix of size $N \times N$ from the distribution of Einsums, we give the following sampling algorithm.

  1. Enumerate triplet factorizations, $T = \{(N_x, N_y, N_z) \in \mathbb{N}^3 : N_x N_y N_z = N\}$. Sample $(N_\alpha, N_\beta, N_\gamma)$ and $(N_\delta, N_\epsilon, N_\phi)$ from $T$. Sampling can be done uniformly or biased to factorizations with more 1s as that induces more varied structures.

  2. Uniformly sample the rank $N_\rho$ from the discrete interval $[1, \min(N_\alpha, N_\epsilon) + 1]$.

  3. Construct tensors $\boldsymbol{X} \in \mathbb{R}^{N_\alpha \times N_\gamma \times N_\delta \times N_\phi \times N_\rho}$, $\boldsymbol{Y} \in \mathbb{R}^{N_\beta \times N_\gamma \times N_\epsilon \times N_\phi \times N_\rho}$ from a standard normal distribution $\mathcal{N}(0, 1)$. Together, these two matrices define a linear operator $\boldsymbol{A}$.

  4. Let $\boldsymbol{I}$ be the identity where $\boldsymbol{I} \in \mathbb{R}^{N \times N}$. Reshape it such that $\boldsymbol{I} \in \mathbb{R}^{N \times N_\alpha N_\beta N_\gamma}$. Finally, compute

  $$\boldsymbol{A}' = \sum_{\alpha\beta\gamma\rho} \boldsymbol{Y}_{\beta\gamma\epsilon\phi\rho} \boldsymbol{X}_{\alpha\gamma\delta\phi\rho} \boldsymbol{I}_{\alpha\beta\gamma} \tag{2}$$

  Now, $\boldsymbol{A}'$ is of shape $(N, N_\delta, N_\epsilon, N_\phi)$. Reshape it to size $N \times N$ to recover $\boldsymbol{A}$.

  In Figure 9 we observe the diverse eigenspectra obtain through the einsum sampling.

- **Diagonal Decays**. To equip the training dataset with a wide variety of eigenspectra, we directly sample various diagonals and then construct matrices from those diagonals. The experiments done in Figure 5 use a uniform mixture over four different diagonal decay distributions. For all of the decays, we evenly space our input values over the $[0, 1]$ interval, i.e. $\{x_i\}_{i=1}^N, x_i = (i-1)/(N-1)$ and sample a scale parameter $s \sim \mathcal{U}[1, 3]$. Each of the following distributions also includes a rate parameter $\alpha$ which controls the rate of spectral decline.

  - **Polynomial**. We sample $\alpha \sim \mathcal{U}[0, 5]$. Then, we compute our eigenvalues $\lambda$ as $\lambda_i = 1 - x_i^\alpha$ and add jitter of $10^{-4}$ to $\lambda_N$ to avoid a zero eigenvalue.

  - **Cosine**. We sample $\alpha$ from a mixture of uniform distributions, $\alpha \sim 0.2 \cdot \mathcal{U}[0.05, 0.1] + 0.2 \cdot \mathcal{U}[1, 4] + 0.6 \cdot \mathcal{U}[0.3, 0.9]$. We then compute, $\lambda_i = 0.5 \cdot (1 + \cos(\pi\alpha x_i))$.

  - **Inverse exponential**. We sample $\alpha \sim \mathcal{U}[0.3, 30]$. Let $f(x) = \frac{1}{1+e^x}$. Then, we compute $\lambda_i = f(\alpha(x_i - 0.5))$.

  - **Log**. We sample $\alpha \sim \mathcal{U}[1, 1000]$ and compute $\lambda_i = 1 - \frac{\log(1+\alpha x_i)}{\log(1+\alpha)}$.

  Given eigenvalues $\lambda$, we can compute $\boldsymbol{D} = s \cdot \text{diag}(\lambda)$. Then, we randomly sample $\boldsymbol{X} \sim \mathcal{N}(0, 1)$ and compute the QR-decomposition $\boldsymbol{QR} = \boldsymbol{X}$. Finally, we compute $\boldsymbol{A} = \boldsymbol{QDQ}^\intercal$ to sample random bases. As $\boldsymbol{Q}$ is an orthonormal matrix, it does not change the eigenvalues present in $\boldsymbol{D}$. In Figure 10 we see the diverse eigenspectra obtained by through our diverse decay functions.

For the $\text{Tr}(\boldsymbol{A})$ and the $\lambda_{\max}(\boldsymbol{A})$ tasks, given a matrix $\boldsymbol{A}$ taken from one of the above distributions, we symmetrize it in order to have real eigenvalues and a well-defined notion of $\lambda_{\max}(\boldsymbol{A})$.

For the $\log|\boldsymbol{A}|$ and $\boldsymbol{A}^{-1}$ tasks, we use positive definite matrices. Specifically, we compute $\boldsymbol{A} \leftarrow \boldsymbol{A}\boldsymbol{A}^\intercal + \epsilon\boldsymbol{I}$. Practically, we use $\epsilon = 10^{-4}$ to ensure that the $\log|\boldsymbol{A}|$ task is well-defined while not being trivial for the transformer model.

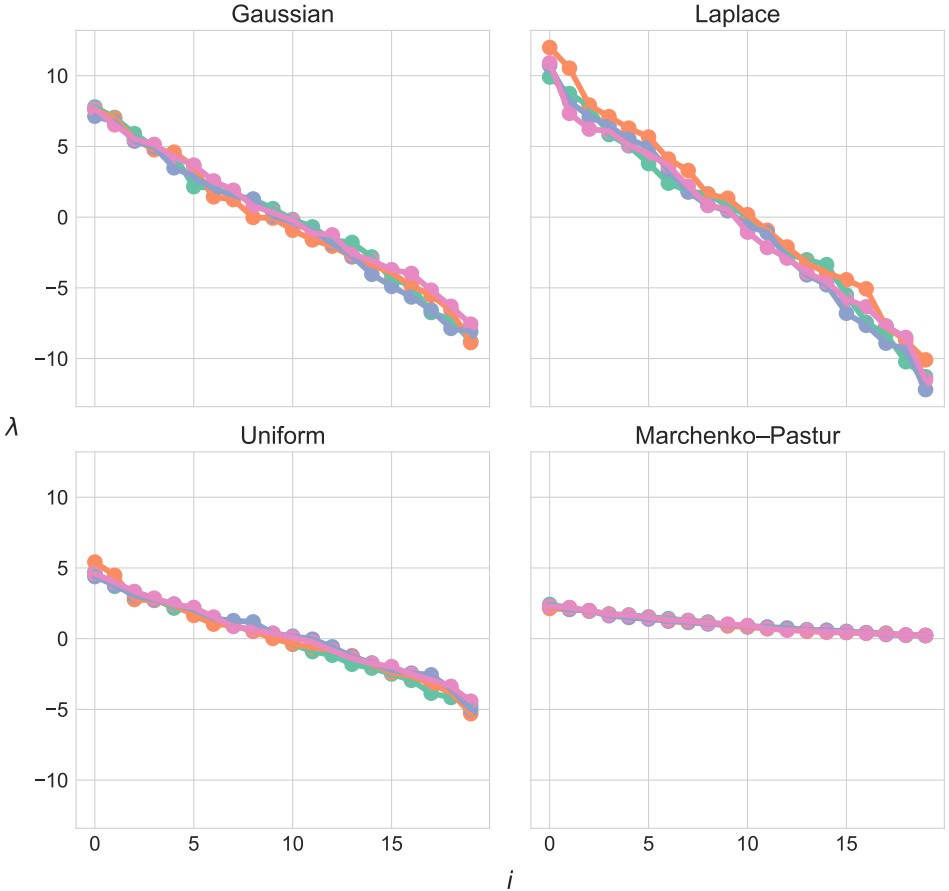

Figure 8: **Eigenspectra of random matrices**. We randomly sample 4 $20 \times 20$ matrices from the selected random matrix families (Gaussian, Laplace, Uniform, and Marchenko-Pastur) and symmetrize them with the exception of Marchenko-Pastur which is symmetric by construction. We plot their eigenvalues and observe that there is very little variance in the eigenspectra within any given random matrix family. We can see how the eigenspectra of each of this families is highly regular and predictible.

## B.2 Matrix Market

To accurately evaluate the performance of our models on OOD matrices, we sample several "real-world", symmetric matrices from the *matrix market* [1] as described in Boisvert et al. (1997). Here are the collections that we used in our experiments:

- BCSSTRUC1. These matrices represent dynamic analyses in structural engineering, containing stiffness and mass matrices.
- BCSSTRUC2. These matrices are linear equation problems "arising from applications of the GT-STRUDL structural engineering code".
- BCSSTRUC3. These matrices were collected from structural engineering packages and serve as generalized symmetric eigenproblems.
- BCSSTRUC4. Symmetric eigenproblems and linear equations.
- CYLSHELL. Finite element discretized octant of a cylindrical shell.
- LANPRO. Lanczos with partial reorthogonalization.
- LAPLACE. Finite different Laplacians.
- PSADMIT. Four symmetric matrices used in modeling power system networks.

All the previous matrices come in a wide variety of shapes. In our experiments, if we need a size of $N \times N$, we simply take the first $N$ columns and the first $N$ rows. That is, we slice the matrix as $A[: N, : N]$ following the notation in `Python`. We also considered projecting them via random matrices. Assuming that $N_0$ is the original size of the matrix, we then have $\boldsymbol{A'} = \boldsymbol{\Omega A \Omega^T}$ where $\boldsymbol{\Omega}$ is a $N_0 \times N$ matrix.

Also, we normalize the matrices by its largest entry as $A_{i,j}/|A_{i^\star,j^\star}|$ where $|A_{i^\star,j^\star}| = \max_{(i,j)}|A_{i,j}|$. We follow this approach for two reasons. First, we avoid having to compute the max eigenvalue or Frobenious norm for each matrix. Second, many of this matrices come in the sparse format, where we only have access to the nonzero entries, and so grabbing the largest one is convenient and fast. This normalization would not make the max eigenvalues approximately 1 but with some divergences as seen in Figure 11.

## B.3 Spectral Analysis

One glaring issue in the usage of random matrices as training data is the homogeneity of their eigenspectra. While previous results such as Charton (2022a) train on up to 360 million random matrices for eigenvalue tasks, there may be limited utility in doing so as almost all of those matrices share very similar spectral properties. In this section, we visually examine the eigenspectra of different distributions of matrices.

Figure 8 visualizes the eigenspectra of different random matrix families. We observe that within any given random matrix family, the eigenspectra follow the same general pattern with very little variation. In fact, the Gaussian, Laplace, and Uniform eigenspectra are all quite similar as well which could explain results like those in Figure 12 where training on those different random matrix families gives very similar results. The fact that the eigenspectra are homogeneous, implies that the neural network can learn that type of pattern and not an algorithm to solve the task at hand.

Figure 9 visualizes the eigenspectra of various Einsums. We observe much greater variation in both the overall shape of the eigenspectra and the values of the largest and smaller eigenvalues, even though the factors of the einsum are also sampled from $\mathcal{N}(0, 1)$.

While Einsums may offer diverse eigenspectra by virtue of representing structured matrices, we can directly control the properties of the eigenspectra by specifying various functional forms. Figure 10 shows the eigenspectra of various diagonal decays which are more varied than both the Einsums and random matrices. By varying the $\alpha$ parameter, we can further control the rate of the decay and the

---

[1] The *matrix market* is provided by the National Institute of Standards and Technology (NIST) as a service, intended for public access and use. See https://www.nist.gov/open/copyright-fair-use-and-licensing-statements-srd-data-software-and-technical-series-publications for further information about licensing.

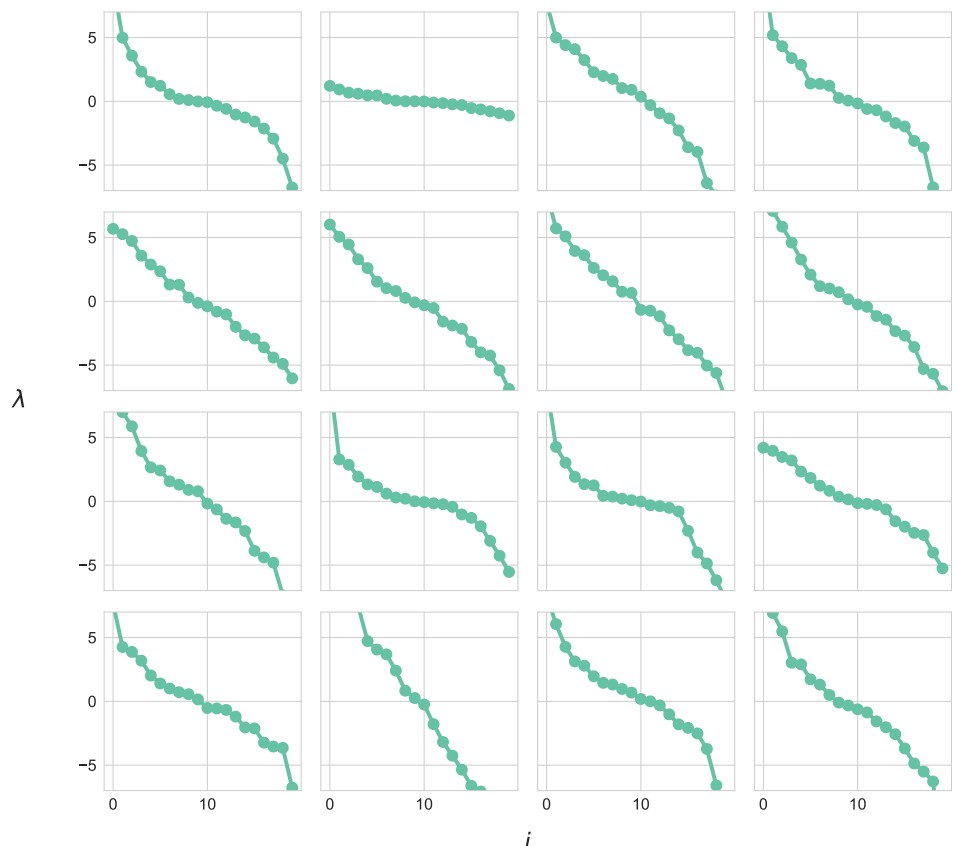

Figure 9: **Eigenspectra of Einsum matrices**. We display the eigenspectra of 16 symmetrized Einsum matrices and observe that the eigenspectra of structured matrices parameterized by Einsums are more varied than the eigenspectra admitted by random matrices.

way in which the eigenspectra decays. Empirically, we observe that this richer distribution results in strong performance improvements (see Figure 5 *(Left)*).

Finally, we visualize the eigenspectra of the *matrix market* matrices in 11 and see that they are qualitatively different from the training distribution making them a good metric for OOD performance. Since these matrices are collected from real-world engineering problems, we observe behavior which is hard to capture through either structured matrices or functional forms such as the step decay in *Bcsstm26* and *Bcsstm02*. In essence, the *matrix market* spectrums are unpredictiable and erratic, making them a great test case for our models.

## C  ARCHITECTURES

In this section we expand on the architectures of `NumFormer` and `RangeFormer` as well as compare them. For a given input $X \in \mathbb{R}^{S \times D}$, where $S$ represents the sequence length and $D$ an embedding dimension, the main components of our core transformer backbone are to apply a positional encoding $X \leftarrow \text{PosEnc}(X)$ and then to iterate $R$ times the following:

$$X \leftarrow X + \text{Attn}(\text{Norm}_A(X))$$
$$X \leftarrow X + \text{MLP}(\text{Norm}_M(X)) \tag{3}$$

Here Norm$(\cdot)$ is a `LayerNorm` layer, Attn$(\cdot)$ refers to a attention mechanism, and MLP$(\cdot)$ is an MLP applied to the embedding dimension. This last MLP consists of two layers with an expansion factor of $4$ similar to (Radford et al., 2018). We discuss below the types of attention mechanisms or MLP activations used. Note that in contrast to equation 1, equation 3 eliminates the dependency on $\ell$, essentially then forcing the model to share weights across the layers.

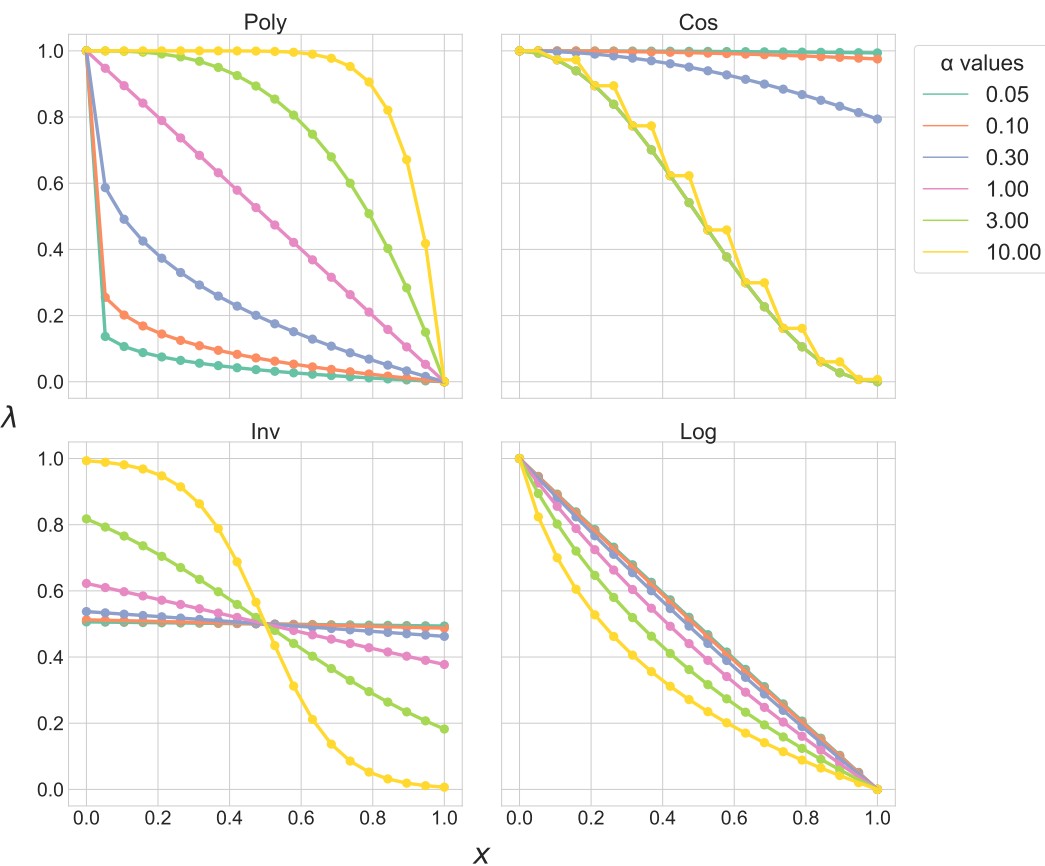

Figure 10: **Eigenspectra of diagonal decays**. The eigenspectra of diagonal decay matrices vary wildly by construction. The parameter $\alpha$ controls the rate and type of decay. For example, with polynomial decay, a large $\alpha$ corresponds to a gentle decline that accelerates later on. With cosine decay, $\alpha$ controls the smoothness of the decline.

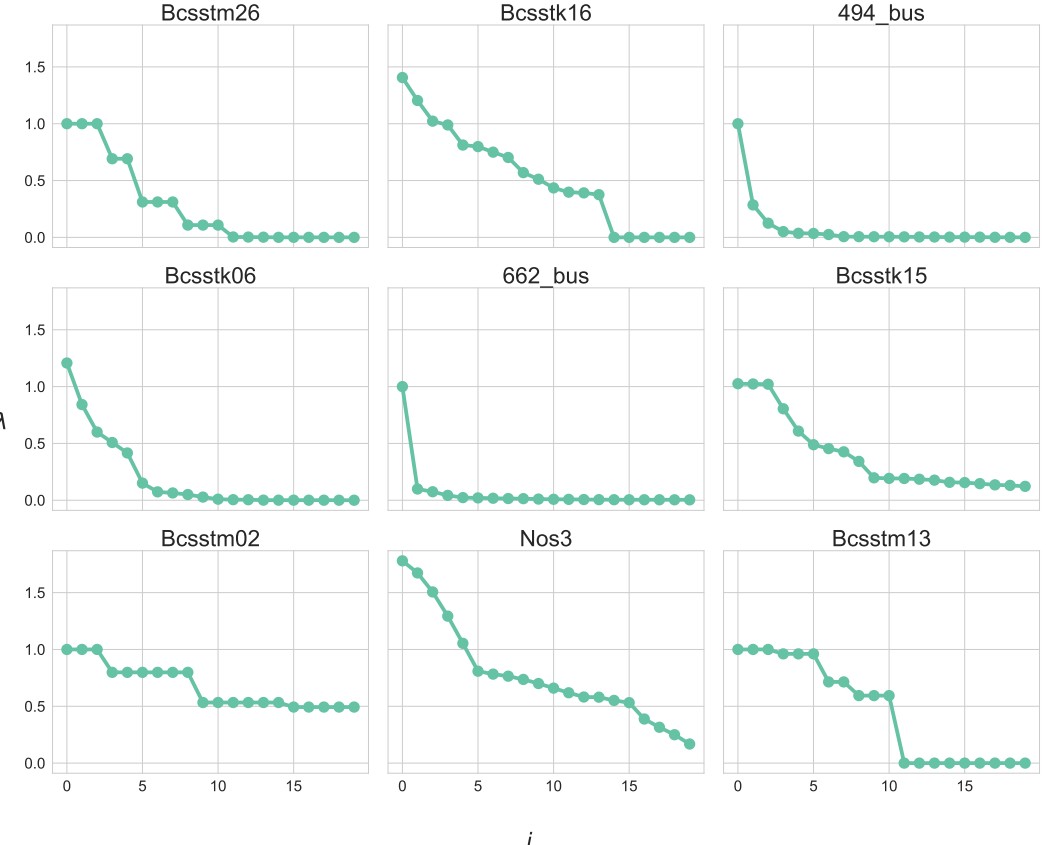

Figure 11: **Eigenspectra of some *matrix market* examples**. The eigenspectra of randomly selected symmetric matrices drawn from structural engineering problems, finite discretizations and other real-world applications are qualitatively different from synthetic random matrix eigenspectra. We take $20 \times 20$ slices from the top left corner of these matrices for this figure and normalize them by their largest element.

| D / N | 20 | 50 | 100 |
|---|---|---|---|
| 8 | $6.6 \times 10^{-1} \pm 2.2 \times 10^{-1}$ | $6.5 \times 10^{-1} \pm 4.3 \times 10^{-1}$ | $8.7 \times 10^{-1} \pm 1.4 \times 10^{-1}$ |
| 32 | $1.7 \times 10^{-1} \pm 3.8 \times 10^{-2}$ | $5.0 \times 10^{-1} \pm 3.6 \times 10^{-1}$ | $4.7 \times 10^{-1} \pm 7.6 \times 10^{-2}$ |
| 128 | $1.9 \times 10^{-1} \pm 2.9 \times 10^{-2}$ | $2.3 \times 10^{-1} \pm 5.3 \times 10^{-2}$ | $2.6 \times 10^{-1} \pm 1.5 \times 10^{-2}$ |
| 512 | $1.8 \times 10^{-1} \pm 1.7 \times 10^{-2}$ | $2.6 \times 10^{-1} \pm 2.4 \times 10^{-2}$ | $2.0 \times 10^{-1} \pm 5.2 \times 10^{-2}$ |
| 1024 | $1.2 \times 10^{-1} \pm 1.1 \times 10^{-1}$ | $2.4 \times 10^{-1} \pm 3.2 \times 10^{-2}$ | $2.4 \times 10^{-1} \pm 4.6 \times 10^{-2}$ |

Table 1: We observe how performance changes as embedding size $D$ and matrix dimension $N$ scale. Small sizes like 8 and 32 limit performance as expected but with $D \geq 128$, performance plateaus. If we suppose that the matrices we work with have an exponentially decaying eigenspectrum or that the majority of eigenvalues are close to 0, then it may be that $D$ does not need to scale with $N$. Fast decaying eigenspectra or small eigenvalues are common characteristics in matrices of interest and is precisely why randomized linear algebra can achieve a good performance with a few vectors even for very large matrices.

| Op | 20 | 100 | 1K |
|---|---|---|---|
| $A^{-1}$ | $1.8 \times 10^{-6}$ | $2.3 \times 10^{-6}$ | $3.1 \times 10^{-4}$ |

Table 2: Error for solves as we increase matrix size. We follow the least square settings from (Liu et al., 2025) and track the MSE error as we increase the problem size.

NumFormer   Given the input $A$, we embed each $A_{i,j}$ into a $D$-dimensional space through a linear layer $W^{(I)} \in \mathbb{R}^{1 \times D}$ to obtain $X \in \mathbb{R}^{N^2 \times D} = \text{vec}(A)W^{(I)}$ and, after passing $X$ through equation 3, we obtain $Y = XW^{(O)} \in \mathbb{R}^{N^2 \times 1}$ through a final linear layer $W^{(O)} \in \mathbb{R}^{D \times 1}$.

For this architecture, $\text{Attn}(\cdot)$ is the standard quadratic attention mechanism (Vaswani et al., 2017) and $\text{MLP}(\cdot)$ uses the standard GeLU activations.

RangeFormer   Given the input $A$, we embed each $A_{i,j}$ into a $D$-dimensional space through a linear layer $\Gamma \in \mathbb{R}^{N \times D}$ to obtain $X \in \mathbb{R}^{N \times D}$ which ultimately capture the range of the operation (where all the action happens). Then we pass $X$ through equation 3 to obtain $Y = XW^{(O)} \in \mathbb{R}^{N \times 1}$ through a final linear layer $W^{(O)} \in \mathbb{R}^{D \times 1}$ which mixes the column dimension.

For this architecture, $\text{Attn}(\cdot)$ is either the linear Taylor attention mechanism (Arora et al., 2023) or the linear polynomial attention mechanism from (Liu et al., 2025). Here for $\text{MLP}(\cdot)$ we use ReLU activations as we found that to be marginally better than GeLU.

However, now the layer $\Gamma$ depends on $N$, which would not allow us to run matrices of different sizes. We have two approaches to circumvent this problem. One approach is to define a $N_{\max}$ (similar to the positional encodings in a transformer that ultimately require a max sequence) and then slice $\Gamma$ to the input size $N \leq N_{\max}$. The other approach is to not make $\Gamma$ learnable but just a random Gaussian embedding. This is an approached inspired by how the randomized linear algebra methods operate. We found that the first approach yields the best results and ultimately used that one in our experiments. Note that this is not a problem as we already have to define a max sequence length for the positional encodings. However, if the user wants to train a model to work for all matrix sizes then then can make $\Gamma$ not learnable.

## D   EXPERIMENTAL DETAILS

### D.1   ADDITIONAL EXPERIMENTS

In this section we expand our statistical analysis to show that the finding from section 3 applies also when the matrices are sampled from a Lapacian distribution, that is $A_{i,j} \sim \mathcal{L}(0, 1)$ or from a uniform on, that is $A_{i,j} \sim \mathcal{U}[-1, 1]$. In Figure 12, we see how, regardless of the sampling distribution, the NumFormer model simply learns the predictible behavior of the matrices and fails to generalize. We also describe experiments on the scaling of the embedding size $D$ with matrix size $N$ as that relationship affects the scalability of our method. Empirically, we do not observe that large $N$ drastically affects performance as long as $D$ is moderately sized.

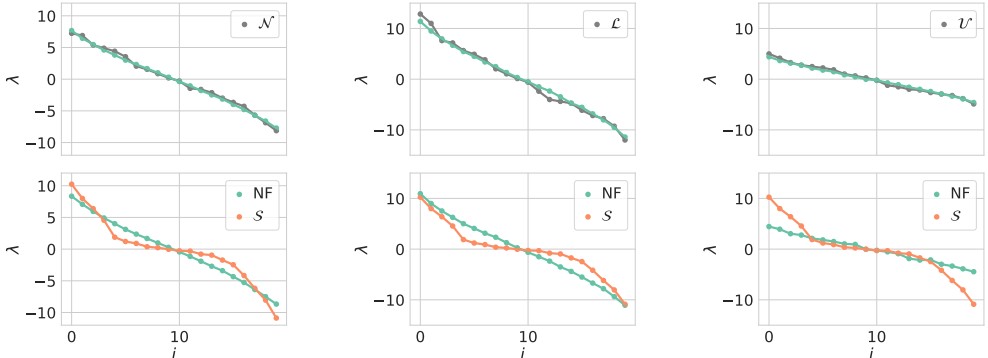

Figure 12: **NumFormer learns particular eigenspectrum decays for different sampling distributions.** In all the top panels, we compare the full eigenspectrum of a random matrix $A$ sampled from different distributions (grey) to the eigenspectrum prediction for the same matrix for a NumFormer model trained on the same distribution of random matrices (green). We see how the NumFormer learns a linear decay in the spectrum that closely matches the training data. In all the bottom panels, we compare the eigenspectrum prediction of a NumFormer but now for a symmetric Toepliz matrix with Gaussian bands $A_{i,j} \sim \mathcal{N}(0,1)$ (orange). We see how the NumFormer fails to predict a different eigenspectrum decay. *Left:* Here $A_{i,j} \sim \mathcal{N}(0,1)$ with $A_{i,j} = A_{j,i}$, that is $A$ is a symmetric Gaussian matrix. *Middle:* Here $A_{i,j} \sim \mathcal{L}(0,1)$ with $A_{i,j} = A_{j,i}$, that is $A$ is a symmetric Laplacian matrix. *Right:* Here $A_{i,j} \sim \mathcal{U}[-1,1]$ with $A_{i,j} = A_{j,i}$, that is $A$ is a symmetric Uniform matrix.

## D.2 FIGURE DETAILS

Here, we provide experimental details for various figures referenced in the paper.

- **Figure 1 Data** We utilize two distinct data distributions - random symmetric $20 \times 20$ Gaussian matrices which represent an easy way to obtain synthetic training data for transformers and diagonal decays which contain more widely varying eigenspectra.
  - **Base Data.** For our baseline Base on the $\mathrm{Tr}(A)$ and $\lambda_{\max}(A)$ tasks, we train using symmetric Gaussian matrices where each element $A_{i,j} \sim \mathcal{N}(0,1)$. For the $\log|A|$ and $A^{-1}$ tasks we use positive definite Gaussian matrices. We sample $A_{i,j} \sim \mathcal{N}(0,1)$ and then compute $A \leftarrow AA^\intercal + \epsilon I$.
  - **Decay Data.** For the Data and Arch. models, we use a uniform mixture over the polynomial, cosine, inverse, and log decay distributions explained in B.1. These matrices are symmetric by construction in the $\mathrm{Tr}(A)$ and $\lambda_{\max}(A)$ tasks. For the $\log|A|$ and $A^{-1}$ tasks, we make these matrices positive-definite in a similar way as to with the baseline Gaussian data.

- **Figure 1 Hyperparameters.** Different tasks perform better with a different selection of hyperparameters for the architecture. For Figure 1 *(c)*, we swept over a set hypers and ultimately used the ones that we detail below. For Figure 1 *(d)*, we did not sweep over hypers, but grab the ones from Figure 1 *(c)* and use them to train models for different sizes. Arguably, larger matrix sizes might require larger models. We now explain: $\mathrm{Tr}(A), \lambda_{\max}(A), \log|A|$, and $A^{-1}$ from Figure 1 *(c)*.
  - $\mathrm{Tr}(A)$. The trace is a control task as we do not usually employ an iterative algorithm to compute the trace. With the NumFormer models we use 2 layers, an embedding size of 256, 8 attention heads, and learned positional encodings. We train for $5,000$ iterations using the AdamW optimizer and a batch size of 100 with a step-wise decay scheduler and an initial learning rate of $10^{-3}$. With the RangeFormer model, we use 8 layers to achieve best performance. Empirically however, we find that we can still out-perform the NumFormer models in Base or Data with only 2 layers.
  - $\lambda_{\max}(A)$. For the the NumFormer model, we use 4 layers, an embedding size of 256, 8 attention heads, and learned positional encodings. For the RangeFormer model, we use the same configuration but with 6 layers instead. Empirically, the RangeFormer performs better with a few more layers while NumFormer performance begins to

deteriorate with more layers. We train using the AdamW optimizer for $10,000$ iterations with a fixed learning rate of $10^{-3}$ and L2 as a loss function. The `RangeFormer` can afford a much larger batch size of $512$ due to its reduced memory footprint while we used a batch size of $128$ for `NumFormer`, else we would run out of memory in our hardware.

- $\log|\boldsymbol{A}|$. For both the `NumFormer` and `RangeFormer` models, we use $4$ layers, an embedding size of $256$, $4$ attention heads, and learned positional encodings. We use a batch size of $128$ and train for $5,000$ iterations using the AdamW optimizer with a learning rate of $10^{-3}$ and a step-wise decay scheduler. For the log-determinant task specifically, we empirically found that the L1 loss function or MAE out-performed standard MSE.

- $\boldsymbol{A}^{-1}$. For the `NumFormer` model, we use $4$ transformer blocks, an embedding size of $256$, $8$ attention heads and learned positional encodings. We train all of the models using a learning rate of $10^{-3}$ and the AdamW optimizer with a step-wise decay scheduler for $10,000$ iterations using a batch size of $128$. For the `RangeFormer` model, we use the same hyperparameters except we use a larger batch size of $256$ as looping reduces the memory footprint of the model. We find this task to be difficult as practically, both the train and test set matrices can be relatively poorly conditioned, on the order of $10^4$ when using $\epsilon = 10^{-4}$. Interestingly, we also find that `RangeFormer` performance does not drop even when we use up to $14$ layers.

- **Figure 5 Data (Left)**. For the Base, Loop, Range, and Attn interventions, we train with random positive-definite Gaussian matrices. The Dist. intervention involves training on a uniform mixture over positive definite decay matrices, explained in Appendix B.1.

- **Figure 5 Hyperparameters (Left).** We fix the hyperparameters over all models to isolate the impact of each individual intervention when learning $\boldsymbol{A}^{-1}$. Specifically, we use $4$ layers, an embedding size of $256$, and $8$ attention heads (although this last change does not apply to the Attn and Dist interventions which use the `BaseConv` alternative to attention). We train for $5,000$ iterations using the nuclear loss $||\mathrm{NN}_\theta(\boldsymbol{A})\boldsymbol{A} - \boldsymbol{I}||_*$ with a batch size of $128$, with the AdamW optimizer, an initial learning rate of $10^{-3}$ and a step-wise decay learning rate scheduler. In the interest of controlling compute, we enforce a limit of GPU time. To get the error bars we loop over $3$ random seeds. In contrast to Figure 1 *(c)* we did not run a sweep over hyperparameters on each intervention. We simply used hypers detailed above and introduce changes one step at a time.

- **Figure 5 Data (Right)**. We use a different training distribution than Figure 5 *(Left)* in order to analyze the impact of training on structured matrices. For the `NumFormer` models, we simply use a uniform mixture of $5 \times 5, 10 \times 10$ and $20 \times 20$ Gaussian random matrices. For the `RangeFormer` models, we similarly use a uniform mixture of $5 \times 5, 10 \times 10$ and $20 \times 20$ random Einsums. For the $\lambda_{\max}(\boldsymbol{A})$ and $\mathrm{Tr}(\boldsymbol{A})$ tasks, we symmetrize these matrices to ensure that we have real eigenvalues. For the $\log|\boldsymbol{A}|$ and $\boldsymbol{A}^{-1}$ tasks, we ensure that these matrices are positive definite. We also observe that for the $\log|\boldsymbol{A}|$ task, additionally training on polynomial decays can help improve performance for the `RangeFormer` method.

- **Figure 5 Hyperparameters (Right)**. We train one model per task that is able to interpolate between observed matrix sizes during training (see appendix C). The `RangeFormer` models all use the `BaseConv` attention alternative, looping, and a learned projection. Empirically, we also find that a L1 loss function may result in marginal performance gains. The `NumFormer` models use standard quadratic attention with $4$ attention heads.

  - $\mathrm{Tr}(\boldsymbol{A})$. For the trace task, the `RangeFormer` uses $4$ layers, an embedding size of $128$ and RoPE. We train for $2000$ iterations using the AdamW optimizer with a step-wise decay scheduler, an initial learning rate of $10^{-3}$, and a batch size of $100$. The `NumFormer` possesses similar hyperparameters but uses only $2$ layers.

  - $\lambda_{\max}(\boldsymbol{A})$. For the maximum eigenvalue task, the `RangeFormer` model uses $4$ layers, an embedding size of $256$ and RoPE. We train for only $1000$ iterations using a batch size of $100$ with the AdamW optimizer, an initial learning rate of $10^{-3}$ and a step-wise decay scheduler. The `NumFormer` possesses similar hyperparameters but uses only $2$ layers.

  - $\log|\boldsymbol{A}|$. For the log-determinant task, the `RangeFormer` model uses $4$ layers, an embedding size of $256$, and RoPE. We train for $5000$ iterations with a batch size of

100, an initial learning rate of $10^{-3}$, and a step-wise decay scheduler. We observe that for the log-determinant task specifically, the `GrokFast` optimizer seems to offer performance improvements albeit inconsistently. The `NumFormer` uses the exact same hyperparameters.

– $\boldsymbol{A}^{-1}$. For the `RangeFormer`, we find that we can attain strong performance with 8 layers, an embedding size of 256, and RoPE. We train for $10,000$ iterations, an initial learning rate of $10^{-3}$, and a batch size of 256. We use the nuclear norm of the difference between $\text{NN}_\theta(\boldsymbol{A})\boldsymbol{A}$ and $\boldsymbol{I}$ as a loss function. For the `NumFormer`, we use the same hyperparameters but with 4 layers and 8 attention heads. We also find that a batch size of 256 practically may substantially slow down training so we recommend a smaller batch size on the order of 128 or 64.

- **Figure 6 (Left)**. Here we use a `RangeFormer` model to fit the $\boldsymbol{A}^{-1}$ task on $50 \times 50$ matrices. For the `RangeFormer`, we use with 8 layers, an embedding size of 256, 8 heads, and learned embeddings. We train for $10,000$ iterations, an initial learning rate of $10^{-3}$, and a batch size of 256. We use the nuclear norm of the difference between $\text{NN}_\theta(\boldsymbol{A})\boldsymbol{A}$ and $\boldsymbol{I}$ as a loss function.

- **Figure 6 (Middle)**. Here we use a `RangeFormer` model to fit the $\text{Tr}(\boldsymbol{A})$ task on varying sizes of matrices: $10 \times 10$, $30 \times 30$ and $50 \times 50$ matrices. For the `RangeFormer`, we use with 4 layers, an embedding size of 128, 4 heads, and learned embeddings. We train for $10,000$ iterations, an initial learning rate of $10^{-3}$, and a batch size of 256. We use the L1 loss.

- **Figure 6 (Right)**. Here we use a `RangeFormer` model to fit the $\boldsymbol{A}^{-1}$ task on $50 \times 50$ matrices. For the `RangeFormer`, we use with 8 layers, an embedding size of 256, 8 heads, and learned embeddings. We train for $10,000$ iterations, an initial learning rate of $10^{-3}$, and a batch size of 256. We use the nuclear norm of the difference between $\text{NN}_\theta(\boldsymbol{A})\boldsymbol{A}$ and $\boldsymbol{I}$ as a loss function.

See Appendix D.3 for ablations on the effect of different positional encodings on our models.

### D.3 POSITIONAL ENCODINGS

In Figure 13, we show the impact of different positional encodings on the performance of `RangeFormer` for the $\text{Tr}(\boldsymbol{A})$ task. Different tasks exhibit different behaviors with the positional encodings. For example, in the inverse task $\boldsymbol{A}^{-1}$, the performance across different positional encodings does not significantly change.

### D.4 WARM-STARTING KRYLOV SOLVERS

In this section we explain how we trained our `RangeFormer` models for this downstream application as well as present some additional results. For both `CG` and `GMRES`, we created a $20 \times 20$ dataset consisting of our spectrum decays. The `RangeFormer` model, for this case, uses Taylor attention, embedding of $D = 128$, looping of $R = 8$ and a step-wise exponentially decreasing scheduler starting the with learning rate of $5 \times 10^{-3}$.

In Figure 14, we provide an initial solution to `CG` or `GMRES` based on our trained `RangeFormer` model. We cap `CG` at 12 iterations and display the convergence behavior. The goal of the experiment is to show how neural network and classical method can interact. As seen in the plots, a neural network trained on many problem could provide some useful information to classical methods. In principle, improvements on the performance of `RangeFormer` would result in faster convergence for the cases shown. Finally, since the *matrix market* contains a wide variety of matrices from diverse applications, it is understandable that our `RangeFormer` models would perform better in some cases and not in others.

### D.5 RNLA

We now explain the details of our `RFSVD` method. For this task, we train a `RangeFormer` model consisting of embedding dimension of $D = 16$, with $H = 1$ number of heads and $L = 2$ layers. However, now the input for the `RangeFormer` model is Gaussian noise $\boldsymbol{\Omega}_0 \sim \mathcal{N}(\boldsymbol{0}, \boldsymbol{I}_{16})$ and the

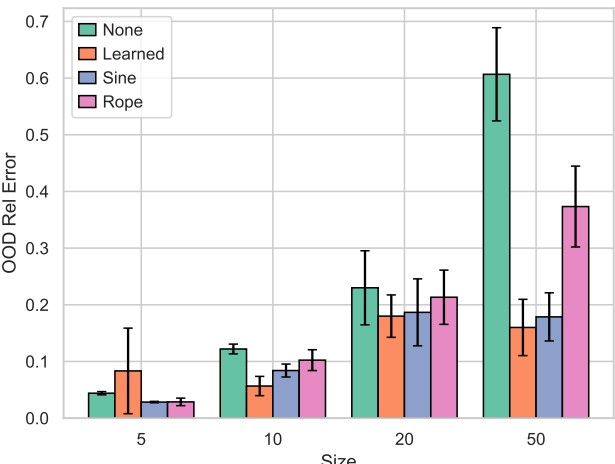

Figure 13: **Performance for different positional encodings**. We observe that the presence of learned, sinusoidal and rotary positional encodings each improve performance on the $\text{Tr}(\boldsymbol{A})$ task for matrices taken from the matrix market. Here, we use `RangeFormer` models with 4 layers and an embedding size of 128. We use the `BaseConv` attention alternative and train for 1000 iterations using the AdamW optimizer with an initial learning rate of $10^{-3}$, a step-wise decay scheduler, a batch size of 128, and MSE as the loss function. Error bars indicate $\pm$ standard deviation around the mean from running 3 different seeds.

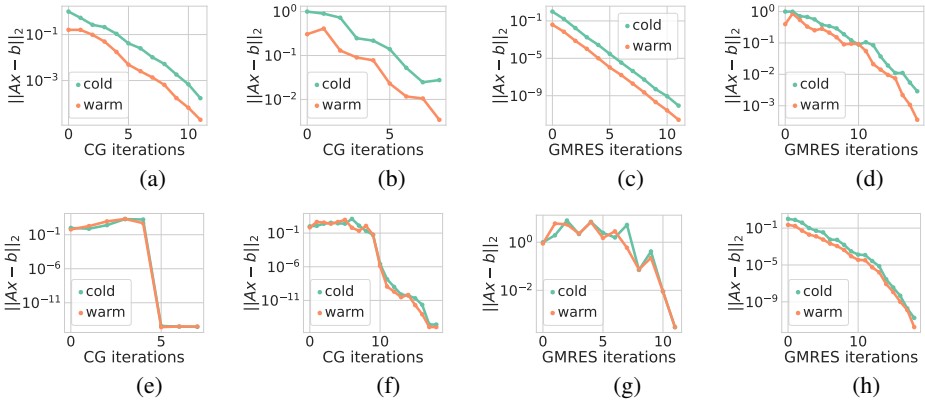

Figure 14: **Warm-start performance for `CG` and `GMRES` on *matrix market*.** For some cases our `RangeFormer` approximation improves converge and, for others, it at least does not harm it. We now detail the matrix used per plot. For *(a)* we used *nos3*, for *(b)* we used *bcsstk09*, for *(c)* we used *gr3030*, for *(d)* we used *bcsstk02*, for *(e)* we used *bcsstm13*, for *(f)* we used *bcsstm06*, for *(g)* we used *bcsstm24*, and, finally, for *(h)* we used *bcsstk15*.

output is $\mathbf{\Omega} = \text{NN}_\theta(\mathbf{\Omega}_0) \in \mathbb{R}^{N \times 16}$. Then $\mathbf{\Omega}$ is fed into the RSVD algorithm and then the approximate spectrum is compared against that of the training matrices (and we backpropagate through the L1 loss). We train the models on $50 \times 50$ matrices using all of our spectral decays.

In Figure 15 we see how we achieve better rank 16 approximations. Additionally, we observe that the learned distribution $\mathbf{\Omega} \sim \text{NN}_\theta$ (Figure 15 (e)) is distinct from Gaussian noise (Figure 15 (d)).

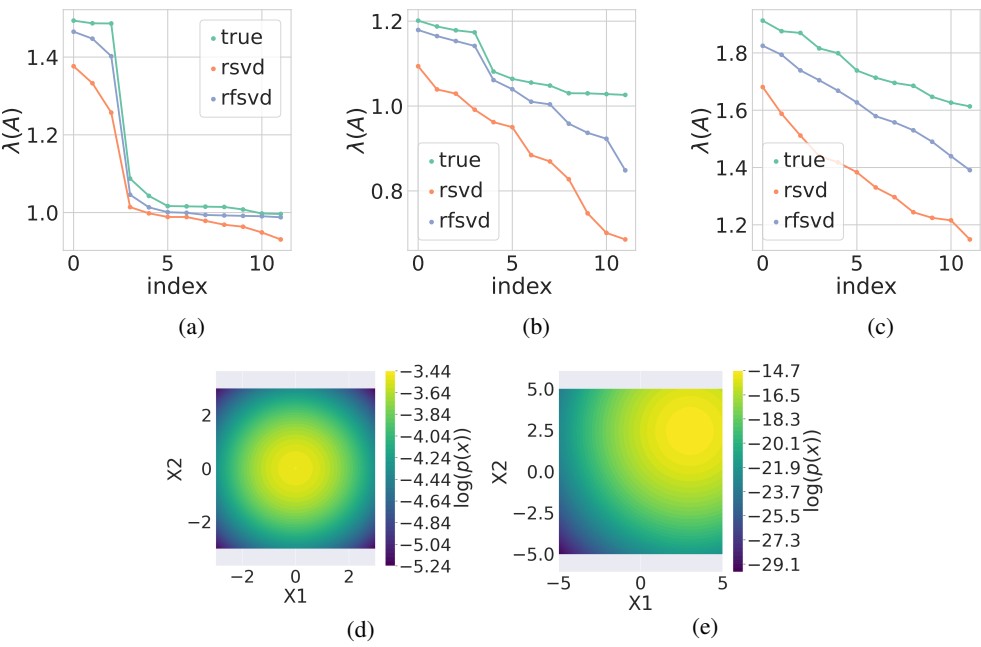

Figure 15: **Impact of learning the sampling distribution.** In *(a)*, *(b)* and *(c)* RFSVD, a RangeFormer model trained to learn a sampling distribution outpeforms RSVD using the same rank of $D = 16$. The matrices per plot are: *(a) 1138 bus*, *(b) bcsstk04*, and, finally, *(c) nos6*. In *(d)*, we show the contour of the first two PCA directions of the sampling distribution for RSVD. As expected, each direction is decoupled. In *(e)*, we show the contour of the first two PCA directions of the learned sampling distribution. Now the directions are coupled.

### D.6 GPs

For this application, we train a RangeFormer model with $H = 8$ heads, $R = 8$ loops, embedding size of $D = 512$, using the AdamW optimizer with learning rate of $10^{-3}$. In contrast to other applications, here we created a specialized training dataset for this application. Let $N$ denote the number of data points used per kernel and $B$ the total number of kernel matrices present in our dataset. For every $b = 1, \ldots, B$, we sample $x_{n,b} \sim \mathcal{U}[0, 10]$, for all $n = 1, \ldots, N$, $\ell_b \sim \mathcal{U}[0, 10]$, and $\sigma_b^2 \sim \mathcal{U}[0.1, 1]$. Then, we construct $\mathbf{K}_b[i, j] = \exp\left(\frac{|x_{i,b} - x_{j,b}|^2}{\ell_b^2}\right) + \sigma_b^2 \delta_{i,j}$ where $\delta_{i,j} = 1$ if $i = j$ and 0 otherwise. The set of kernels $(\mathbf{K}_b)_{b=1}^B$, where each $\mathbf{K}_b \in \mathbb{R}^{N \times N}$, constitutes our training dataset. Indeed, the motivation for the training dataset is to ensure that the RangeFormer model performs on a wide range of RBF kernels, where the range is motivated by the bounds in the sampling uniform distributions above.

For the experiment discussed in Figure 7 *(Right)*, we follow a similar procedure as above but we only create one kernel $\mathbf{K}_1 \in \mathbb{R}^{N \times N}$ ($B = 1$) and then we sample $y_n \sim \mathcal{N}(\mathbf{0}, \mathbf{K}_1)$ for $n = 1, \ldots, N$. After that, then we provide the data set $(x_n, y_n)_{n=1}^N$ to then find $\phi = (\ell, \sigma^2)$ that minimized the loss $\mathcal{L}(\phi) = \log |\mathbf{K}_\phi| + \mathbf{y}^T \mathbf{K}_\phi \mathbf{y}$ discussed in section 5.3. We then compared the train hyperparameters $\ell^\star$ to then ones sampled $\ell_1$ (Figure 7 *(Right)*). This procedure is a common check done in probabilistic modeling, as we want to ensure that our training procedure can recover the parameters of the data generating process.

For the experiment discussed at the end of section 5.3, we follow the same procedure from the previous paragraph but now with the data $(x_n, y_n)_{n=1}^N$ where $x_n \sim \mathcal{U}[0, 1]$, $\epsilon_n \sim \mathcal{N}(0, 0.04)$ and $y_n = \sin(2\pi x_n) + \epsilon_n$ for $n = 1, \ldots, N$. In contrast to the previous experiments, there is no ground truth $\ell_1$ that we should recover. However, we note that the learned hypers $\ell_{\text{RF}}^\star$ are approximately the same as those that would be learned when using `Cholesky` $\ell_{\text{Chol}}^\star$, that is $\ell_{\text{RF}}^\star \approx \ell_{\text{Chol}}^\star$.

Overall, the main goal of these experiments is to test how would a `RangeFormer` model would perform when substituting linear algebra routines, not to advocate that GP kernels should now be trained with transformers. It is a challenging task, at at every iteration $t$, the hypers change $\phi^{(t)}$ and so now the `RangeFormer` model has to perform linear algebra operations against $\boldsymbol{K}_{\phi^{(t)}}$ and, worse, the trajectory of $\phi^{(t)}$ is determined by how accurate the linear algebra operations compute the loss.

### D.7 Loss Details

The different tasks illustrated in 5 require a variety of loss functions and model configurations. Basically, we have two sets of losses. The losses that apply to vectors of size $N$ or scalars and the losses that apply to matrices of size $N \times N$. For the first category we have the L1 loss $\|\boldsymbol{x} - \boldsymbol{y}\|_1 = \sum_i |x_i - y_i|$ and the L2 loss $\|\boldsymbol{x} - \boldsymbol{y}\|_2 = \sqrt{\sum_i (x_i - y_i)^2}$, which if required we could report as $\|\boldsymbol{x} - \boldsymbol{y}\|_2^2$. We mostly use the L1 as a default as we find that to be better performing, though marginally, than the L2 loss. We use the L1 loss for the following tasks: $\boldsymbol{A}^{-1}\boldsymbol{b} \in \mathbb{R}^N$, $\log |\boldsymbol{A}|$, $\lambda(\boldsymbol{A}) \in \mathbb{R}^N$, $\lambda_{\max}(\boldsymbol{A}) \in \mathbb{R}$, and, $\text{Tr}(\boldsymbol{A})$.

For the matrices, we use the nuclear norm $\|\boldsymbol{A}\|_* = \sum_i \sigma_i$ where $\sigma_i$ are the singular values of $\boldsymbol{A}$. In our experiments, we only use matrix losses for the inverse task, and actually, our loss for this task would take the form of $\|\text{NN}_\theta(\boldsymbol{A})\boldsymbol{A} - \boldsymbol{I}\|_*$. Intuitively, we are asking that the neural network approximation $\text{NN}_\theta(\boldsymbol{A})$ *behaves* like the identity but not that it matches its entries $\|\text{NN}_\theta(\boldsymbol{A}) - \boldsymbol{A}^{-1}\|_*$. Our choice of loss also has the added benefit of not requiring to compute the inverse $\boldsymbol{A}^{-1}$ for the training task.

## E Runtime Analysis

We provide a formal analysis of the time complexity of a forward pass through a single attention head in a vanilla transformer (`NumFormer`) against the proposed `RangeFormer`.

For the following analysis, we denote $\boldsymbol{W}^{(I)}$ as the input projection layer, and $\boldsymbol{W}^{\boldsymbol{Q}(1)}, \boldsymbol{W}^{\boldsymbol{K}(1)}, \boldsymbol{W}^{\boldsymbol{V}(1)}$ as the query, key, and value projections. Here, the 1 indicates that we do this analysis for a single attention head. Additionally, let $D$ be the embedding dimension of the model.

Suppose we have some input matrix $\boldsymbol{A} \in \mathbb{R}^{N \times N}$. The `NumFormer` then flattens $\boldsymbol{A}$ into a sequence vector of length $N^2$. Each element of the sequence $A_{i,j}$ is embedded into a $D$-dimensional space resulting in $\boldsymbol{X} \in \mathbb{R}^{N^2 \times D} = \text{vec}(\boldsymbol{A})\boldsymbol{W}^{(I)}$.

A single attention head then constructs $\boldsymbol{Q}^{(1)} = \boldsymbol{X}\boldsymbol{W}^{\boldsymbol{Q}(1)}, \boldsymbol{K}^{(1)} = \boldsymbol{X}\boldsymbol{W}^{\boldsymbol{K}(1)}$ and $\boldsymbol{V}^{(1)} = \boldsymbol{X}\boldsymbol{W}^{\boldsymbol{V}(1)}$ where $\boldsymbol{W}^{\boldsymbol{Q}(1)}, \boldsymbol{W}^{\boldsymbol{K}(1)}, \boldsymbol{W}^{\boldsymbol{V}(1)} \in \mathbb{R}^{D \times D}$. Thus, the construction of $\boldsymbol{Q}^{(1)}, \boldsymbol{K}^{(1)}, \boldsymbol{V}^{(1)} \in \mathbb{R}^{N^2 \times D}$ runs in $\mathcal{O}(N^2 D^2)$

The attention matrix is then constructed as $\sigma(\boldsymbol{Q}^{(1)}\boldsymbol{K}^{(1)\mathsf{T}})\boldsymbol{V}^{(1)}$ where $\sigma$ is the softmax function. The matrix multiplication $\boldsymbol{Q}^{(1)}\boldsymbol{K}^{(1)\mathsf{T}}$ runs in $\mathcal{O}(N^4 D)$ and constructs a matrix of size $N^2 \times N^2$, incurring a space complexity of $\mathcal{O}(N^4)$. The final matrix-multiply with $\boldsymbol{V}^{(1)}$ incurs the same time complexity so the total time complexity of a single forward pass of an attention head with batch size 1 is $\mathcal{O}(N^4 D + N^2 D^2)$.

The major change proposed by the `RangeFormer` architecture is to avoid flattening the matrix and instead directly embed the range of the operator through a learnable projection. Given $\boldsymbol{A} \in \mathbb{R}^{N \times N}$, `RangeFormer` computes $\boldsymbol{X} \in \mathbb{R}^{N \times D} = \boldsymbol{A}\boldsymbol{W}^{(I)}$.

The construction of the key, query, and value matrices is functionally the same but the sequence length is now $N$ instead of $N^2$. Thus, $\boldsymbol{Q}^{(1)}, \boldsymbol{K}^{(1)}$ and $\boldsymbol{V}^{(1)} \in \mathbb{R}^{N \times D}$ incur a time complexity of $\mathcal{O}(ND^2)$.

If a practitioner chooses to use `RangeFormer` with the standard attention mechanism, then the total time complexity of a forward pass for one attention head would be $\mathcal{O}(N^2D + ND^2)$ with a space complexity of $\mathcal{O}(N^2)$. However, replacing the standard attention mechanism with a subquadratic alternative can offer an even more scalable architecture. We review two alternatives: linear attention and `BaseConv`.

Linear attention replaces the softmax function $\sigma$ with two feature maps $\sigma'$ separately applied to the key and query matrices. Concretely,

$$\text{LinAttn}(\boldsymbol{Q}^{(1)}, \boldsymbol{K}^{(1)}, \boldsymbol{V}^{(1)}) = \sigma'(\boldsymbol{Q}^{(1)})\sigma'(\boldsymbol{K}^{(1)\intercal})\boldsymbol{V}^{(1)} \tag{4}$$

The first matrix multiplication $\sigma'(\boldsymbol{K}^{(1)\intercal})\boldsymbol{V}^{(1)}$ has a time complexity of $\mathcal{O}(ND^2)$ resulting in a matrix of size $D \times D$. The additional matrix multiply against $\boldsymbol{Q}^{(1)}$ incurs the same time complexity resulting in a total time complexity of $\mathcal{O}(ND^2)$.

Another subquadratic alternative to the attention mechanism is `BaseConv` introduced in Arora et al. (2023), meant to simulate gating and convolutions. Where $\boldsymbol{X}^{(l)} \in \mathbb{R}^{N \times D}$ is the output of the $l$'th layer of the transformer, we compute linear projections.

$$\boldsymbol{U}^{(A)} = \boldsymbol{X}\boldsymbol{W}^{(A)} \tag{5}$$

$$\boldsymbol{U}^{(B)} = \boldsymbol{X}\boldsymbol{W}^{(B)} \tag{6}$$

Where $\boldsymbol{K}$ represents the kernel of a 1-dimensional convolution, we compute the depth-wise convolution:

$$\boldsymbol{U}^{\text{conv}} = \boldsymbol{K} * \boldsymbol{U}^{(A)} \tag{7}$$

Finally, we compute:

$$\boldsymbol{X}^{(l+1)} = \boldsymbol{U}^{\text{conv}} \odot \boldsymbol{U}^{(B)} \tag{8}$$

Equations 5 and 6 run in $\mathcal{O}(ND^2)$ resulting in the same time complexity as linear attention.

# F    CHARACTERIZING THE EFFECTIVENESS OF RANGEFORMER

The core idea behind the `RangeFormer` is to capture most of the action of a matrix through a learnable low-rank projection matrix. We expect the low-rank projection to work as there exist theoretical guarantees for using random Gaussian matrices as projection matrices. Below, we reproduce the problem statement and proof from **?**, Theorem 11.5.

Let $\boldsymbol{A} \in \mathbb{R}^{n \times n}$ be an input matrix with $l < n$ be the subspace matrix. The *rangefinder* problem is to find an orthonormal matrix $\boldsymbol{Q} \in \mathbb{R}^{n \times l}$ whose range aligns with the largest left singular vectors of $\boldsymbol{A}$. We define the spectral norm error as

$$||\boldsymbol{A} - \boldsymbol{Q}\boldsymbol{Q}^*\boldsymbol{A}|| = ||(\boldsymbol{I} - \boldsymbol{Q}\boldsymbol{Q}^*)\boldsymbol{A}|| \tag{9}$$

The rank-$l$ matrix $\hat{\boldsymbol{A}} = \boldsymbol{Q}\boldsymbol{Q}^*\boldsymbol{A}$ serves as an approximation for $\boldsymbol{A}$.

Fix $\boldsymbol{A} \in \mathbb{R}^{n \times n}$ with singular values $\sigma_1 \geq \sigma_2 \geq \ldots$. Draw a standard normal test matrix $\boldsymbol{\Omega} \in \mathbb{R}^{n \times l}$ and construct $\boldsymbol{Y} = \boldsymbol{A}\boldsymbol{\Omega}$. Choose $k < l - 1$ and introduce the random variable $Z = ||\boldsymbol{\Gamma}^\dagger||$ where $\boldsymbol{\Gamma} \in \mathbb{R}^{k \times l}$ is also standard normal. Define $\boldsymbol{P_Y}$ as the orthogonal projector onto the range of $\boldsymbol{Y}$. Then,

$$\mathbb{E}||(\boldsymbol{I} - \boldsymbol{P_Y})\boldsymbol{A}|| \leq \left(1 + \sqrt{\frac{k}{l-k-1}}\right)\sigma_{k+1} + (\mathbb{E}Z)\left(\sum_{j>k}\sigma_j^2\right)^{1/2} \tag{10}$$

The randomized rangefinder finds an $l$-dimensional subspace that captures as much of the action of $\boldsymbol{A}$ as the best $k$-dimensional subspace.

*Proof.* Since $\boldsymbol{\Omega}_{i,j} \sim \mathcal{N}(0, 1)$, the matrices $\boldsymbol{\Omega}_1 := \boldsymbol{V}_1^*\boldsymbol{\Omega}$, $\boldsymbol{\Omega}_2 := \boldsymbol{V}_2^*\boldsymbol{\Omega}$ are also independent standard normal since $\boldsymbol{V}_1, \boldsymbol{V}_2$ are orthonormal and mutually orthogonal. Then by Chevet's Theorem referenced in **?**, Prop 10.1,

$$\mathbb{E}||\boldsymbol{\Sigma_2}\boldsymbol{\Omega_2}\boldsymbol{\Omega_1}^\dagger|| = \mathbb{E}_{\Omega_1}\mathbb{E}_{\Omega_2}||\boldsymbol{\Sigma_2}\boldsymbol{\Omega_2}\boldsymbol{\Omega_1}^\dagger|| \tag{11}$$

$$\leq \mathbb{E}[||\boldsymbol{\Sigma_2}||_F||\boldsymbol{\Omega_1}^\dagger|| + ||\boldsymbol{\Sigma_2}||||\boldsymbol{\Omega_1}^\dagger||_F] \tag{12}$$

$$\leq \sqrt{\frac{k}{l-k-1}}||\boldsymbol{\Sigma_2}|| + (\mathbb{E}Z)||\boldsymbol{\Sigma_2}||_F \tag{13}$$

The last inequality involves a well-known estimate of the trace of an inverted Wishart matrix described in **?** Prop 10.2.

We can insert estimates of the expectation for $Z$. When $2 \leq k < 1$, $\mathbb{E}Z \leq \frac{e\sqrt{l}}{l-k}$. When $k \ll l$, $\mathbb{E}Z \approx \frac{1}{\sqrt{l} - \sqrt{k}}$. These estimates lead to accurate performance bounds across many matrices and parameters.

For `RangeFormer` specifically, we would expect our method to perform better for matrices with rapidly declining eigenspectra as that way, the action of the matrix can be better captured in a low-rank projection. For example, in Figure 10, using a polynomial diagonal decay, we would not expect our method to work so well for $\alpha = 10$ as it has many large eigenvalues without decay but would expect `RangeFormer` to perform better for $\alpha \in [0.05, 0.3]$. Similarly, we would expect performance improvement on matrices in the matrix market shown in Figure 11 due to their declining eigenspectra.

