# OpenReview forum: "Understanding and Relaxing the Limitations of Transformers for Linear Algebra"
_ICLR.cc/2026/Conference — ICLR 2026 Poster_

### Official Review · Reviewer_gjAT · 2025-10-26

**Soundness:** 3
**Presentation:** 3
**Contribution:** 2
**Rating:** 8
**Confidence:** 2

**Summary:**

The goal of the paper is to study the capability of the transformer architecture to perform numerical linear algebra tasks, such as linear system solving, computing SVD and EVD and so on. The first contribution of the paper is that the existing architecture approaches is unable to learn generalizable algorithms and show OOD generalization. Their main contribution is RangeFormer, which has various features such as learnable matrix embeddings, linear attention, and a diversifying the training distribution. These improvements first  reduce the make the transformer efficient for linear algebra tasks and also improves OOD generalization.
They show their new architecture can scale well compared to the classical transformer in performing various tasks such as Gaussian process learning and randomized numerical linear algebra. Furthermore the paper also explores using transformers to improve the performance of classical methods such as Krylov subspace methods and randomized SVD by providing better initialization and sampling distributions.

**Strengths:**

The main strength of the paper is in using linear algebraic tasks as a benchmark for understanding the limitations of transformer architecture. Various works have shown such limitations for different algorithmic tasks, and this paper delves deeper into tasks pertaining to implementing algorithms for numerical linear algebra.

**Weaknesses:**

Perhaps a minor weakness of the paper is in terms of the writing, and I would suggest the authors to include a main summary of their key findings in the beginning of the paper.

**Questions:**

I found the paper quite interesting and I have a few questions regarding it. Firstly there has been quite a lot of work on linear algebraic tasks such as linear regression and the ability of transformers to solve them in context ( for eg. refer to this work https://arxiv.org/abs/2211.15661). As far as I understand your current work pretrains the architecture for linear algebraic tasks, however it would be interesting to know if claims along the lines that the transformer has enough expressivity to perform tasks such as SVD or linear system solving in context would be quite interesting.

---

> ### Author Response · Authors · 2025-11-21
>
> We greatly appreciate your thoughtful and supportive review! We also value your acknowledgement that studying the limitations of the transformer architecture for basic mathematical operations is an important contribution for improving the capabilities of this architecture.
>
> Indeed, an interesting next step in our work is to train RangeFormer models to solve linear algebra operations in-context. Currently, we have to train one model per task, but arguably if we train in-context models then we could have a unified architecture for all tasks, where the transformer knows which task is being required by the examples that it is being passed. It is only our new scalable RangeFormer architecture which now makes in-context feasible as the memory requirements are much lower therefore allowing us to pass multiple matrix examples as a possibility. Concretely, if our RangeFormer model could learn an embedding similar to an eigendecomposition of each matrix, then that embedding would serve both to compute several linear operations like solves, trace, log det and clearly SVD. We believe that in-context training might be a sensible suggestion to encourage our RangeFormer models to learn such widely useful embeddings and we will explore this avenue as future work!
>
> Moreover, in the new PDF we have added a summary of key findings at the end of the introduction as per your suggestion. Based on your review we see that you have a good understanding of our paper. We would greatly appreciate it if you could also address some misunderstandings that other reviewers might have during the discussion period.

---

### Official Review · Reviewer_w4zs · 2025-10-28

**Soundness:** 3
**Presentation:** 2
**Contribution:** 2
**Rating:** 4
**Confidence:** 3

**Summary:**

This paper performs an empirical analysis of transformers trained in a supervised manner to perform linear algebra tasks. The authors design detailed experiments to identify the bottlenecks of transformers in performing such operations. Specifically, they find that transformers tend to learn properties of the input data distribution rather than the underlying matrix operation algorithms.
To address this, the authors propose a new training recipe and a novel combination of architectural components to mitigate the identified limitations. Finally, they evaluate the improved architecture on three downstream applications.

**Strengths:**

* The paper conducts a detailed and comprehensive analysis of the limitations of transformer-based architectures in performing matrix operations. It further proposes a new model architecture and training recipe, and evaluates them on a wide range of tasks to demonstrate their effectiveness.

* The paper provides thorough details of all experimental settings, which is good for future reproducibility.

* The work uses simple linear algebra tasks as a testbed to reveal that transformers may primarily learn the statistical properties of the data distribution rather than the underlying algorithmic procedures.

**Weaknesses:**

* Overall, the paper is a solid empirical study. However, my main concern is in the motivation for investigating the transformer’s ability to perform such simple linear algebra tasks.
As the authors repeatedly emphasize, (e.g., line 86: “We note that the goal of our work is in understanding the potential capabilities and limitations of transformers for linear algebra…”; line 472: “This research area is in its early stages… our goal is not to compete with purpose-built classical algorithms…”),  the stated goal is primarily exploratory. However, it remains unclear why understanding transformers’ capabilities on these tasks is important, and what future directions this understanding enables. Does this line of research aim to eventually develop a single transformer that outperforms classical linear algebra solvers? Why is a new transformer architecture needed to solve problems that classical solvers already handle perfectly? It is also not evident whether the insights obtained from these experiments will translate to a better understanding of transformer usefulness in real-world analytical settings.

* Many experiments in the main text lack multi-run results with different random seeds. While the appendix provides details about whether hyperparameter optimization was performed, the plots in the main text appear to come from single runs. Including multiple seeds would strengthen the statistical reliability of the results. Since the models are relatively small and trained for fewer than 10,000 iterations, adding multi-run averages and std should be feasible.


* Some aspects of the writing logic could be improved. For example, in line 297 -line 300, the authors state "We now construct, step-by-step, our RangeFormer architecture, and introduce our synthetic data procedure....", but the following paragraph immediately shifts to discussing experimental results instead of the mentioned architecture.

**Questions:**

*  I appreciate the detailed and carefully designed experiments examining the transformer’s limitations in solving linear algebra tasks. However, as mentioned in the weakness section, I would appreciate a more in-depth explanation of the importance and broader motivation behind this problem. Specifically, what kind of future research directions or practical applications does understanding transformers’ ability to perform linear algebra enable?

*  Could the authors clarify why they selected the three specific downstream tasks in Section 5? Are these tasks directly related to the failure cases identified in Section 3, or do they represent scenarios where classical solvers perform poorly or are computationally expensive, thus providing opportunities where RangeFormer offers advantages?

---

> ### Author Response · Authors · 2025-11-21
>
> We appreciate your thoughtful review and we now address your feedback. Let us elaborate on the motivation of our work. Our motivation is really threefold.
>
> First, matrix operations are a fundamental primitive for many machine learning systems. Therefore if we are to continue to pursue transformers as a backbone for general-purpose systems (as the community is doing), then it will be important for them to have some basic competence with matrix operations. Perhaps an analogy might be helpful. Humans are nowhere near as good as calculators at addition and multiplication. Yet it would be likely a bad idea to prescribe that we don’t learn competency at arithmetic, and instead rely on calculators, from a young age. That basic competency is important for performance in all sorts of related tasks. As a fundamental primitive, good performance on matrix operations is likely to transfer to a range of important abilities. For this reason, we do not believe it would be fair to summarily reject any work developing transformers for matrix operations, just as it would be unfortunate if any work developing transformers for arithmetic were rejected out of hand. Transformers are indisputably of great and broad interest, and applied very widely, and so it is of scientific interest to understand their abilities and limitations on key mathematical operations.
>
> Second, by investigating the limitations of transformers for matrix operations, we can gain some broader insights into learning behaviour. For example, we uncovered that prior work for transformers on matrix operations is performing statistical interpolation, failing on even trivially OOD matrices, such as the identity matrix. Through a series of interventions, we can move more towards algorithm discovery, with good performance in OOD settings. This finding has broader implications about OOD generalization, especially given the generality of matrix operations.
>
> Thirdly, and finally, we do hope to lay the groundwork for learning systems that could either overtake or complement classical algorithms for matrix operations. Classical solvers are certainly well-developed and mature, but they do not handle matrix operations perfectly.  Much like neural graph approaches initially were no competition for classical methods, after several years, they have become practically significant, and yielded breakthroughs like solving a conjecture in Kazhdan-Lusztig polynomials [1, 2]. Similarly, in this instance, it is not reasonable to expect that one paper is going to make transformers better than decades of work on purpose-built classical solvers, representing millions of human hours of effort. But, it is crucial that we invest in the fundamental research that can make this outcome eventually possible. Another helpful analogy could be in differential equations. Initially, classical solvers vastly outperformed neural PDE methods, and still do in several settings. However, the field of neural networks for PDE is nascent, and operating on quite different principles than classical solvers. Now, these methods tend to outperform classical methods in higher dimensional settings, and where smoothness of the solution can provide a good prior. It didn’t take one paper — this is a finding that is emerging after years of foundational work.

---

> > ### Author Response · Authors · 2025-11-21
> >
> > In terms of other questions:
> >
> >
> > We have made our runs with multiple seeds, usually 3 seeds for each hyperparameter combination Figure 5 (left) but the error bars are extremely small and therefore unnoticeable.
> >
> > We selected those three specific downstream applications mainly because of their widespread use (linear solvers and RSVD are workhorses), their use of many linear algebra operations (solves, eig, log-det) and because we are familiar with these applications. It is worth noting that our work has significantly raised the standard when compared to prior methods which could not be used for real problems or in real ML pipelines.
> >
> > Finally, we hope you can consider raising your score in light of our clarifications, especially regarding our motivation. There’s an ongoing debate of course about whether we should give transformers access to tools or whether we should develop approaches to provide end-to-end solutions to various problems. In different settings, each side of this debate can have some merit. But we do think that work on one side should not be summarily dismissed. Moreover, we genuinely believe our work has broad foundational significance, and is worthy of support. It’s not possible to go from the beginning steps to a system that will overtake a mature area of research in a single paper, or even a single year. But it is worth investing in alternative approaches to important problems of broad significance. As we have outlined above, our contributions are also multifaceted, and provide insights into the behaviour of transformer models in being able to perform OOD generalization.
> >
> > _References_
> >
> > [1] Blundell et al., 2021. Towards combinatorial invariance for Kazhdan-Lusztig polynomials. arXiv 2111.15161
> >
> > [2] Davies et al., 2021. Advancing mathematics by guiding human intuition with AI. Nature.

---

> > > ### Comment · Reviewer_w4zs · 2025-11-28
> > > **Score Update and Other Suggestions**
> > >
> > > I thank the reviewer for the detailed clarification. My original concern on the motivation and impact has been addressed. Since motivation was my primary reservation, I will raise my score from 4 to 6 accordingly.
> > >
> > > Minor suggestion:
> > >
> > > * I appreciate the clarification on the multi-seed experiments. It might help to state this explicitly in the paper and provide a quantitative example showing how small the std are across experiments (Figures 1–7). This will make the results more convincing.
> > >
> > > * A related-work section would also strengthen the paper. It might help readers  in related research fields distinctguish the contribution the paper provides comparing to previous works. For example, currently many researches work on the transformer capability for learning algorithm both in-context and supervised (e.g., works cited at `line 171`), or ability in arithmetic tasks [1,2]. A brief discussion of these directions could help distinguish this paper’s unique perspective. For example, if existing work already shows transformers learning machine learning algorithms such as least squares (which inherently involve linear algebra), what additional insights does this paper contribute on top of those results?
> > >
> > > * The abbreviation NLA (numerical linear algebra) is used before being defined.
> > >
> > > * The code link provided at `line 72` in the revision points to an expired repo.
> > >
> > > ---
> > >
> > > [1] McLeish, Sean, et al. "Transformers can do arithmetic with the right embeddings." NeurIPS (2024)
> > >
> > > [2] Lutz, Patrick, et al. "Linear Transformers Implicitly Discover Unified Numerical Algorithms." ICLR (2024).

---

### Official Review · Reviewer_XjAK · 2025-10-28

**Soundness:** 3
**Presentation:** 3
**Contribution:** 2
**Rating:** 2
**Confidence:** 3

**Summary:**

This paper presents a comprehensive empirical investigation into the capability of transformer architectures to execute fundamental linear algebraic operations. Through their simulations, the authors assess whether transformers can accurately perform tasks such as vector addition, matrix multiplication, and eigenvalue estimation. The results provide empirical evidence that, despite their expressive power, transformers—particularly those constructed following existing theoretical proofs—struggle to reliably perform even simple linear algebra operations.
Finally, the paper proposes a new version of transformer that is capable of performing those tasks.

**Strengths:**

* The simulation section is comprehensive.
* The research question this paper tries to address is indeed new to the ML community.

**Weaknesses:**

My main concern of this paper is that I do not see how the results from this paper would provide insights to either empirical or theoretical studies of transformer.
I believe a main statement this paper tries to make is that the transformer constructed in [Garg et al., 2023, Von Oswald et a. 2023], does not appears to be the same as the learned transformer.
However, this does not seem to be a significant insight to the theoretical community.
It is obvious that in a series of works on "transformers simulate algorithms", their proofs are mainly constructive, meaning they only show the existence of a solution instead of the uniqueness of it.
Therefore, while analyzing transformers block-by-block seems to be a new approach to my knowledge, I think it is obvious that trained transformers will not necessarily converge the the exact construction from previous studies.


My second concern is that even from the empirical perspective, I am unsure what does the insight or the proposed model from this paper contribute to empirical ML.
I understand that this paper is not trying to push SOTA performance. However, if ones goal is to utilize transformers for linear algebra operations, why not simply follow the constructions in previous studies?

**Questions:**

1. Can the authors explain why would adding an extra matrix embedding layer would help the performance? At least in the linear regression task [Garg et al., 2023, Von Oswald et a. 2023], multiplying the data matrix with a random matrix results in another linear regression problem. Therefore, intuitively I do not understand how would it work.

2. Can the authors address my two concerns? It is possible that I misunderstand some of the goals of this paper.

3. One observation from the paper is that trained transformers do not have good OOD generalization on linear algebra tasks, which seems to be an universal issue other than for this specific task. Would training transformers on a more diverse dataset gain performance?

---

> ### Author Response · Authors · 2025-11-21
>
> We appreciate your feedback. We would like to clarify some fundamental misunderstandings.
>
> Our work provides several insights for improving the numerical capabilities of transformers. For example, we showed that prior methods [1,2,3] are essentially broken as they cannot scale for matrices beyond 30 x 30 and catastrophically fail with slight deviations from the training data such as the identity matrix (section 3) because they are just learning statistics of the data (Figure 3). In contrast, we showed that our models can generalize to real OOD matrices from the _matrix market_ benchmark through two interventions. First, by training with synthetic data that is sufficiently varied in terms of matrix structures and eigenspectra. Interestingly, as seen in Appendix B, the spectrum of the benchmark matrices is distinctively different from our synthetic matrices, but training with a wider range promotes more generalizable behavior in our models. Second, we incorporate diverse architectural modifications like random range projection, looping and polynomial attention. Our novel use of range projection allows us to scale up to 1K x 1K matrices whereas prior methods could only do up to 30 x 30 and this intervention is motivated in the new Appendix F [4].
>
> Regarding your comment on prior work, we clarify that our work addresses a different problem than Garg et al 2023 and Von Oswald et al 2023. Those works study in-context learning: the transformer receives examples $(x1, w^T x1, x2, w^T x2, …, x_{q})$ and must make a prediction for $w^Tx_{q}$. In contrast, we study approximating matrix functions: given matrix $A$, output $f(A)$. As we are addressing a fundamentally different problem, we make no claims about converging to their theoretical construction. Additionally, while multiplying the input matrix with a random embedding matrix in a linear regression task just amounts to another linear regression task in those prior works, it serves as a form of _dimensionality reduction_ in our work. The projection serves to embed the input matrix into the residual stream of the transformer. This is an alternative to flattening the matrix which improves the scalability of the RangeFormer. In the new Appendix F, motivated by [4] we show that the random projection preserves the eigenspectra of our matrix, making the problem tractable.
>
> In summary, we significantly enhance the constructions from previous studies, for scalability and performance improvements by _orders of magnitude_. We also provide fundamental insights into the limitations of past studies, especially on out-of-distribution matrices. Finally, we show promising empirical results on downstream tasks where repeated matrix operations are required, including combinations with classical solvers which outperform classical solvers alone.
>
> In light of these clarifications, we would appreciate it if you could reconsider your score.
>
> _References_
>
> [1] Charton, 2022. Linear algebra with transformers. Transactions on Machine Learning Research.
>
> [2] Charton, 2022. What is my math transformer doing? - Three results on interpretability and generalization. arXiv:2211.00170.
>
> [3] Liu et al., 2025. Towards Learning High-Precision Least Squares Algorithms with Sequence Models. ICLR.
>
> [4] Martinsson, P. and Tropp, J. Randomized Numerical Linear Algebra: Foundations & Algorithms. arXiv 2022.01387. 2020.3

---

### Official Review · Reviewer_Qhzi · 2025-10-30

**Soundness:** 3
**Presentation:** 3
**Contribution:** 3
**Rating:** 6
**Confidence:** 2

**Summary:**

This paper begins by identifying the inherent limitations and failure modes of Transformer-based approaches to matrix computation. The authors’ analysis reveals that standard Transformers primarily act as statistical interpolators rather than genuine algorithm discoverers, which leads to particularly poor performance on out-of-distribution (OOD) data. Building upon these insights, the paper introduces a series of architectural improvements aimed at enhancing both scalability and accuracy, culminating in a new model termed RangeFormer. The proposed architecture demonstrates substantially improved scalability and performance on OOD matrices from the Matrix Market collection. Moreover, the paper provides, to the best of the authors’ knowledge, the first empirical evidence that Transformers can be effectively applied to downstream tasks involving iterative matrix operations, such as Gaussian process learning and improving sampling distributions in randomized numerical methods.

**Strengths:**

* The paper introduces a learnable projection mechanism that embeds the input matrix into a higher-dimensional latent space. This design substantially improves computational efficiency and scalability, while enhancing the Transformer’s performance on matrix-multiplication–type tasks.

* By adopting a looped Transformer structure inspired by iterative methods in numerical linear algebra, the model can repeatedly process the same matrix multiple times. This enables adaptive refinement and better handling of tasks with varying difficulty and condition numbers.

* The authors construct a novel structured data mixture for training, consisting of matrices with diverse structures and eigenspectrum decay patterns. Compared to conventional Gaussian-random training data, this approach significantly improves out-of-distribution (OOD) generalization.

* The paper provides the first empirical demonstration that Transformers can be successfully applied to downstream tasks involving iterative matrix operations, such as Gaussian process learning and improving sampling distributions in randomized numerical methods.

**Weaknesses:**

I will admit I am not very familiar with this paper, so I am basing my review more on the clarity of the writing.


1. The theoretical explanation of the RangeFormer design remains largely intuitive. The paper proposes four key modifications, range embedding, looping, linear attention, and structured data mixture, but their effectiveness is demonstrated only empirically, without rigorous theoretical justification or detailed analysis. In particular:
* Why can the projection operator $X = A\Gamma$ preserve the matrix spectrum or operator properties?
* Must the projection be Gaussian? In practice, Gaussian projections are dense and computationally expensive, could sparse alternatives such as CountSketch or other structured embeddings achieve similar effects?
* Why does *linear attention* improve the approximation of matrix multiplications?
* What are the convergence and stability guarantees of the *looping* mechanism?
* Does the structured data mixture genuinely improve algorithmic generalization, or does it merely enlarge the training support domain?

2. RangeFormer is derived from the baseline NumFormer through several architectural modifications. While the paper describes these individual changes, it does not provide a complete framework diagram of RangeFormer that would help readers clearly understand the overall workflow and methodological pipeline of the proposed model.

**Questions:**

* The notation in the figures (e.g., “HP”) is insufficiently explained, making them difficult to interpret.
* It is recommended that the main text include additional references to the literature on RandNLA and RSVD.
* In the comparison between RSVD and RFSVD, the paper only reports the error metrics but does not provide a comparison of the running time. Although RFSVD improves accuracy, it remains unclear whether this improvement comes at the cost of increased computational time.

---

> ### Author Response · Authors · 2025-11-21
>
> We appreciate your thoughtful review and feedback.
>
> Regarding the theoretical explanation of our work. We know from [1] that looping transformers _can_ represent linear algebra programs as primitives. However, the question is now how do we actually train transformers with such capabilities in practice. Our work provides a clear demonstration of the type of data (different structured and eigenspectra) that the models need to be trained on to learn meaningful algorithms that generalize in practice. Additionally, our choice of having our transformer models operate on the range of the matrix and not on its entries comes from the field of randomized linear algebra (RNLA). The key insight that most of the algorithms in RNLA leverage is that we can understand several of the properties of a matrix by looking at how it operates on a handful of random complementary directions. In the new version of the paper you can find a section in the new Appendix F “Characterizing the Effectiveness of RangeFormer”.
>
> Specifically, with regards to your questions:
>
> The projection operator $X = A \Gamma$ can intuitively maintain the operator properties with high probability because the random projection is of full rank (all directions are independent). We’ve added Appendix F providing a formal proof of the above statement. The projection also does not _need_ to be Gaussian - they were initially chosen as they are easy to sample. Based on your suggestion, we ran experiments with structured embedding matrices, namely Sparse Sign and Random Trigonometric embeddings (see Ch. 9 of [3]). For computing the maximum eigenvalue, Sparse Sign achieves a OOD relative error of 0.396 on the matrix market,  Random Trigonometric 0.305 and our original approach 0.25 on computing the max eigenvalue. It does seem that we can choose cheaper embedding methods but at a mild performance cost. This is a valuable insight especially for working with larger scale matrices and we appreciate the suggestion!
>
> Linear attention is free from scale normalization activations like the standard softmax which are not favorable to approximating primitives like matrix multiplies. Section 3.3 of [2] provides a more thorough justification.
>
> Like most deep learning setups, we do not have a specific theoretical convergence guarantee for the looping mechanism. However, we can easily monitor the performance of looping by querying how the model is solving the linear algebra problem and stop accordingly. For example, in linear solves, we can always check the residual $||A x^{\star} - b|| $ where $x^{\star} = NN(A)$.
>
> We firmly believe that we are improving algorithmic generalization because we are evaluating our model on 50 hard real-life benchmark matrices from the matrix market whose idiosyncratic spectrum (Figure 11) is not easily captured by the smooth spectra that we used in our training data (Figure 10). We believe that once the spectra of the input matrix is sufficiently diverse then the transformer models start to learn something useful, rather than we need to train with data that is quite similar to our benchmarks.
>
> We believe our work is making a foundational contribution, both in advancing transformers for fundamental operations, but also in understanding their key limitations, and how we can move away from statistical interpolation and towards algorithm discovery. We hope you can consider raising your score in light of our clarifications, and the timeliness and significance of this work and its multifaceted contributions.
>
> _References_
>
> [1] Giannou et al., 2023. Looped Transformers as Programmable Computers. arXiv 2301.13196.
>
> [2] Liu et al., 2025. Towards Learning High-Precision Least Squares Algorithms with Sequence Models. ICLR.
>
> [3] Martinsson, P. and Tropp, J. Randomized Numerical Linear Algebra: Foundations & Algorithms. arXiv 2022.01387. 2020.

---

### Official Review · Reviewer_LWQ3 · 2025-11-06

**Soundness:** 2
**Presentation:** 3
**Contribution:** 1
**Rating:** 2
**Confidence:** 4

**Summary:**

Starting with the argument that a "generally intelligent system" should be perfectly competent at performing matrix operations, this work studies Transformers' ability to perform primitives relevant to linear algebra (e.g., eigenvalue computations, trace computation, etc.). Results show the Transformers architecture explored in this work is unable to perform well, but interventions on the architecture based on seen failures leads to a new architecture---called rangeformer---which can fix these issues. In essence, rangeformer is a series of interventions involving changes from positional encodings to weight tying (aka looping) to data level interventions.

**Strengths:**

The paper's experiments are well written and the sequential structure made it fun to follow.

**Weaknesses:**

- My core apprehension is that I do not currently see sufficient novelty or utility to the paper. The paper is not necessarily a science of deep / machine learning project that tries to elicit new phenomena or explain known ones (i.e., does not ask question of the sorts "why does a Transformer fail to perform a task"), nor is it an explicit methodological improvement (i.e., it does not try to introduce a new architecture for practical tasks and verify the efficacy of proposal). This, personally, makes it hard for me to contextualize this paper with respect to existing literature.

- Related to above, if I focus on the precise narrative the authors took for motivating their study, i.e., generally intelligent systems should be able to perform linear algebra, then I find myself not convinced at all by this narrative. For several problems that Transformers were benchmarked on in this paper, I argue I would personally fail to solve several of these tasks with a sufficiently high performance. That said, I can easily write a program or do a library call to perform these deterministic calculations, and so can Transformers. In this sense, I don't really see the motivation to wanting to replace a deterministic calculator with a fuzzy one.

  - To be clear, I'd generally deem a point like this to be minor and not flag it in the review. However, given that this statement was made multiple times by the authors and because I do not see a sufficient motivation for the paper, I'm leaning into authors' proposed motivation.

- Prior work: Authors state (L46--48) that there is little prior work exploring how Transformers can perform linear algebra tasks. I strongly disagree with this. There's several papers exploring, e.g., the ability of a model to perform tasks like matrix completion [1], broader math operations [2], identifying the benefits of looping for this [3], effects of data properties [4], and so on.

[1] https://arxiv.org/abs/2410.22244

[2] https://arxiv.org/abs/2506.13688

[3] https://arxiv.org/abs/2502.17416

[4] https://arxiv.org/abs/2307.03381

**Questions:**

See weaknesses.

---

> ### Author Response · Authors · 2025-11-21
>
> While we appreciate your feedback, we must respectfully disagree with the points that you raised in your review. Prior work [1,2,3,4,5,6] has argued that “Transformers can do linear algebra” but in our work we show that this statement is not accurate. As seen in section 3, prior methods are unable to scale to even tiny matrices larger than 30 x 30, and unable to handle even trivial matrices outside of the training distribution, such as the identity matrix. Instead, we show in Figure 3 that prior methods are simply learning statistics about their training data but not solving the underlying linear algebra problem. Section 3 both addresses your question about the utility of the work as well as providing an explanation for the failure cases.
>
> In terms of novelty, prior work [1,2,3,4,5,6] applied a transformer to a matrix by taking each entry as part of a sequence but this approach is not scalable as attention would require $N^4$ memory (Figure 1(d)) where $N \times N$ is the size of the matrix. In contrast, we are the first to show that instead of having the model identify patterns based on the entries of a matrix, the model should rather identify patterns based on how the matrix acts on a random subspace, which is also a key insight in the field of randomized linear algebra [7].
>
> Let us elaborate on the motivation of our work, which is multifaceted. You will agree that matrix operations are ubiquitous in machine learning systems. A general-purpose system should therefore have basic competence. We would never prescribe that people stop learning basic arithmetic simply because they could have a calculator at their disposal. Competence in basic mathematical primitives is essential to being able to competently approach a wide range of tasks, even if specialized tools can outperform us in certain instances. Moreover, these competencies are related to other competencies which we cannot necessarily anticipate: having competence at arithmetic, and matrix operations, is going to translate to many other related tasks. It would not be reasonable to summarily reject any paper developing transformers for basic mathematical primitives, simply because specialized tools exist for certain operations.
>
> Secondly, and crucially, our motivation is also to help lay the groundwork for learning systems that could either overtake or complement classical algorithms for matrix operations. Much like neural graph approaches initially were no competition for classical methods, after several years, they have become practically significant, and yielded breakthroughs like solving a conjecture in Kazhdan-Lusztig polynomials [8, 9]. Similarly, in this instance, it is not reasonable to expect that one paper is going to make transformers better than decades of work on purpose-built classical solvers, representing millions of human hours of effort. But, it is crucial that we invest in the fundamental research that can make this outcome eventually possible. Neural networks for PDEs is another similar instance: while initially they did not outperform classical solvers, they are beginning to provide superior performance, especially in higher dimensional settings. For fundamental operations, it’s important to invest in a diversity of approaches, which will ultimately have complimentary advantages. We provide the first demonstrations of success for transformers performing matrix operations in _downstream_ tasks, including hybrid approaches involving classical methods.
>
> _References_
>
> [1] Charton, 2022. Linear algebra with transformers. Transactions on Machine Learning Research.
>
> [2] Charton, 2022. What is my math transformer doing? - Three results on interpretability and generalization. arXiv:2211.00170.
>
> [3] Liu et al., 2025. Towards Learning High-Precision Least Squares Algorithms with Sequence Models. ICLR.
>
> [4] Garg et al., 2023. What Can Transformers Learn In-Context? A Case Study of Simple Function Classes.
>
> [5] Giannou et al., 2023. Looped Transformers as Programmable Computers. arXiv 2301.13196.
>
> [6] Yang et al., 2024. Looped Transformers are Better at Learning Learning Algorithms. ICLR.
>
> [7] Martinsson, P. and Tropp, J. Randomized Numerical Linear Algebra: Foundations & Algorithms. arXiv 2022.01387. 2020.3
>
> [8] Blundell et al., 2021. Towards combinatorial invariance for Kazhdan-Lusztig polynomials. arXiv 2111.15161
>
> [9] Davies et al., 2021. Advancing mathematics by guiding human intuition with AI. Nature.

---

### Meta-Review · Area_Chair_rA9i · 2026-01-07

**Summary:**

The article studies transformers ability to implement basic linear algebra operations, points at failure modes, and proposes interventions.

Strengths include comprehensive analysis of limitations, proposed beneficial alterations, and clarity of the chosen linear algebra framework. Concerns include lack of novelty and utility in theory and practice for real world tasks as well as disagreements with the motivation. A favorable review has low confidence.  Another reviewer raised their score from 4 to 6.

In spite of valid criticism, the article contributes valuable steps in a relevant direction. Although a more extensive discussion period would have been beneficial, my impression is that the strengths outweigh the criticism. Therefore I am recommending accept.

**Reviewer Concerns:**

Reviewer LWQ3:
I do not currently see sufficient novelty or utility to the paper
I strongly disagree with ``little prior work exploring how Transformers can perform linear algebra tasks.''
-> Response in rebuttal


Reviewer Qhzi:
does not provide a complete framework diagram of RangeFormer that would help readers clearly understand the overall workflow
-> This weakness seems to have remained uncommented

Reviewer XjAK:
I do not see how the results from this paper would provide insights to either empirical or theoretical studies of transformer.
-> Response in rebuttal

**Reviewer Scores:**

For each review, specify how you think the reviewer would have changed their score if they had been able to participate fully in the discussion.

Reviewer LWQ3: 2 -> 2 / 4
Reviewer Qhzi: 6 -> 6 (not very familiar)
Reviewer XjAK: 2 -> 2 / 4
Reviewer w4zs: 4 -> 6
Reviewer gjAT: 8 -> 8 (low confidence)

---

### Decision · Program_Chairs · 2026-01-26

Accept (Poster)